# Wrapped $\beta$-Gaussians with compact support for exact probabilistic modeling on manifolds

**Sergey Troshin**                                                    *s.troshin@uva.nl*
*Language Technology Lab, Informatics Institute, University of Amsterdam*

**Vlad Niculae**                                                      *v.niculae@uva.nl*
*Language Technology Lab, Informatics Institute, University of Amsterdam*

**Reviewed on OpenReview:** *https://openreview.net/forum?id=KrequDpWzt*

## Abstract

We introduce wrapped $\beta$-Gaussians, a family of wrapped distributions on tractable Riemannian manifolds, supporting efficient reparametrized sampling, as well as exact density assessment, effortlessly supporting high dimensions and anisotropic scale parameters. We extend Fenchel-Young losses for geometry-aware learning with wrapped $\beta$-Gaussians, and demonstrate the efficacy of our proposed family in a suite of experiments on hypersphere and rotation manifolds: data fitting, hierarchy encoding, generative modeling with variational autoencoders, and multilingual word embedding alignment.

The source code is available at https://github.com/ltl-uva/wbg.

## 1 Introduction

The Euclidean space is not always sufficiently expressive for modeling the rich data encountered in applied machine learning. Sometimes data naturally lives in another geometry. Examples include geological processes (Curray, 1956) or word embeddings (Meng et al., 2019) where data is represented on spheres. Hierarchical data such as knowledge graphs or taxonomies may be better represented in hyperbolic spaces (Nickel & Kiela, 2017), while the manifolds of rotations and of rigid motions are used in applications like robot pose or motion estimation (Rosen et al., 2019) or as domains for parametrizing other models (Artetxe et al., 2016).

Probabilistic modeling on Riemannian manifolds is a research area attracting increasing attention. Geometry introduces substantial computational challenges compared to the Euclidean case. For instance, whereas the multivariate Gaussian distribution allows efficient calculations over $\mathbb{R}^n$ for any symmetric positive semidefinite covariance matrix, the Gaussian has no direct equivalence over a Riemannian manifold that is as efficient, exact, and expressive. The possible extensions, further described in §2.2, either require numerical approximations, numerical integration, rejection sampling, sacrifice expressivity of the scale parameter, or abandon geometry in favor of working in a restriction of the ambient Euclidean space.

A promising direction is that of *wrapped distributions* (Chevallier & Guigui, 2020), defined implicitly in terms of a core distribution in tangent space. Wrapped distributions support efficient sampling by design, but their exact density is in general intractable. Unless the manifold has non-positive curvature everywhere, like hyperbolic geometry (Nagano et al., 2019), in general, any point on the manifold can be reached by the wrapping of infinitely many tangent points (Figure 1, left). In order to have exact expressions, it is, therefore, necessary to use a compactly-supported yet expressive distribution in the tangent space, yet no such constructions have been proposed so far.

We fill this gap with $\beta$-Gaussians (Martins et al., 2022), a recent family generalizing Gaussians, which provide controllable compact support, tractable sampling, and natural loss functions. Our main contributions are:

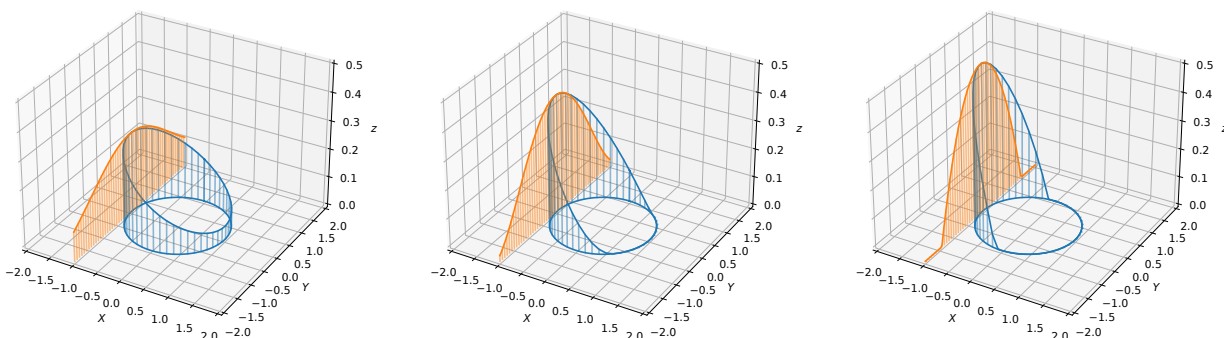

**Figure 1:** Wrapped $\beta$-Gaussian on $\mathbb{S}^1$ with (a) $\beta = 1$ (wrapped Gaussian), (b) $\beta = 0.5$ and (c) $\beta = 0$. The $\beta$-Gaussian (orange), defined over the tangent plane at $\mu = (-1, 0)$, is wrapped over the circle $x^2 + y^2 = 1$ to obtain the wrapped $\beta$-Gaussian distribution (blue). In (b,c), wrapping is injective on the support.

- the wrapped $\beta$-Gaussian, a distribution family on Riemannian manifolds, providing exact density assessment and reparametrized sampling even in high dimensions and with anisotropic scale;

- an efficient embedded parametrization of scale matrices for the case of hyperspheres;

- geometry-aware Fenchel-Young losses for parameter fitting;

- experimental demonstration of the usefulness of wrapped $\beta$-Gaussians in synthetic and real modeling tasks on hyperspherical $\mathbb{S}^n$ and special orthogonal $SO(n)$ manifolds (§4), including hierarchy modeling, variational autoencoders, and multilingual word embedding alignment tasks, in high dimensions.

The wrapped $\beta$-Gaussian distribution is applicable whenever a standard toolbox is available and tractable for the manifold: exponential and logarithmic mapping, the determinant of their Jacobian, and the injectivity radius; a fairly broad (but not universal) class of manifolds. Particularly, it includes the class of symmetric Riemannian manifolds (Chevallier et al., 2022), including hyperspheres, hyperbolic spaces, and matrix manifolds such as $SO(n)$ and $SE(n)$.

Wrapped $\beta$-Gaussians tend toward wrapped Gaussians in the limit of $\beta \to 1$, and the proposed losses tend toward log-probability and Kullback-Leibler divergences, but for any $\beta < 1$ the support of the distribution is compact, allowing for efficient and exact computation, even on manifolds with finite injectivity radius.

## 2 Background

### 2.1 Differential geometry

A smooth manifold $\mathcal{M}$ is a topological space that locally resembles Euclidean space. Common examples include the sphere, the torus, the real projective plane, as well as various matrix manifolds such as the special orthogonal group. The tangent spaces allow to generalize calculus to manifolds: at each point $x \in \mathcal{M}$, the tangent space at $x$, denoted $\mathcal{T}_x \mathcal{M}$, is a real vector space isomorphic to $\mathbb{R}^d$, where $d$ is the dimension of the manifold. A Riemannian manifold is a manifold in which every tangent space $\mathcal{T}_x \mathcal{M}$ is equipped with an inner product $g_x$, and thus also with an induced norm $\|v\|_x$.

Every tangent vector uniquely defines a geodesic. The exponential mapping $\mathrm{Exp}_x : \mathcal{T}_x \mathcal{M} \to \mathcal{M}$ maps a tangent vector $v$ to a point on the manifold by traveling along the corresponding geodesic for a unit of time (thus $\|v\|$ controls the velocity). The exponential mapping is not injective in general. For instance, on the circle $\mathbb{S}^1$, two tangent vectors $v$ and $2\pi + v$ map to the same point: a unit-time journey along the geodesic defined by $2\pi + v$ means travelling in the same direction as $v$ except faster by $2\pi$, and so will wrap around and end up at the same destination. However, $\mathrm{Exp}_x$ is injective on a small enough ball $B(0, r) = \{v \in \mathcal{T}_x \mathcal{M} : \|v\|_x \leq r\}$.

The **injectivity radius** of $\mathcal{M}$ at $x$ is the radius of the largest such ball:

$$\mathrm{inj}_x \mathcal{M} \coloneqq \sup\{r > 0 : \mathrm{Exp}_x \text{ is injective on } B(0, r)\}.$$

The canonical mapping in the opposite direction is known as the logarithmic map. If we denote by $\mathrm{Exp}_x^{-1}$ the set-valued inverse (preimage), then $\mathrm{Log}_x : \mathcal{M} \to \mathcal{T}_x\mathcal{M}$ is defined as[1]

$$\mathrm{Log}_x(y) \coloneqq \arg\min\{\|v\|_x : v \in \mathrm{Exp}_x^{-1}(y)\}.$$

When the minimizer above is not unique, (*e.g.*, for the antipodal point on a circle), $\mathrm{Log}_x$ is undefined. With these definitions, on the set $B(0, \mathrm{inj}_x \mathcal{M}) \subseteq \mathcal{T}_x\mathcal{M}$, $\mathrm{Exp}_x$ and $\mathrm{Log}_x$ form a pair of inverse diffeomorphisms.

Finally, we discuss the special case of *embedded manifolds*, which covers all manifolds used in our experiments. In many cases of interest a manifold can be represented as a subset of $\mathbb{R}^n$. In this case, the tangent space can also be taken as a subset of $\mathbb{R}^n$, specifically $\mathcal{T}_x\mathcal{M} = \{v \in \mathbb{R}^n : \langle v, x \rangle = 0\}$, and the inner product and Riemannian metric are inherited from $\mathbb{R}^n$.

## 2.2 Statistics on manifolds

Standard probability and statistics on Euclidean vector spaces enjoy the benefit of linearity which are no longer available when generalizing to manifolds. In particular, on manifolds, there is no unique generalization of the Gaussian distribution. We briefly present the few alternatives and their tradeoffs.

**The intrinsic approach (Pennec, 2006).** A natural and principled way to define probability distributions on manifolds is in terms of the intrinsic geodesic distances $d(x, y)$. The intrinsic standard Gaussian centered at $\mu$) takes the form $p(x) \propto \exp(-d^2(x, \mu)/2)$, with its extension to full scale matrices:

$$p(x) \propto \exp\left(-\frac{1}{2}\mathrm{Log}_\mu(x)^\top \Sigma^{-1} \mathrm{Log}_\mu(x)\right). \tag{1}$$

This formulation can be derived from a maximum entropy principle on the manifold, but it generally does not lead to computationally friendly reparametrized sampling, and while unnormalized densities are tractable, the normalization constant might not be (see Hauberg (2018) for the instantiation on $\mathbb{S}^n$, where the normalizing constant is only available in the isotropic case ).

**The embedded approach.** For manifolds embedded in a vector space $\mathbb{R}^n$, the embedded approach relies on defining a distribution in the ambient space and conditioning (renormalizing) it on the manifold. This approach is sometimes computationally friendlier: in fact, conditioned on $\mathbb{S}^n$, $p(x) \propto \exp(-\kappa\|x - \mu\|^2)$ yields the celebrated Langevin distribution, also known as von Mises–Fisher (Mardia & Jupp, 1999, §9.3.2). Still, more complex anisotropic constructions are generally intractable. Moreover, since distances in ambient space are not necessarily related to distances on the manifold, this approach may misrepresent the geometry.

**The wrapped approach (Mardia & Jupp, 1999, §3.5.7).** The wrapped distribution has a long history in directional statistics (Mardia & Jupp, 1999) with most early instances targeting the circle (Stephens, 1963). Since the tangent space of a Riemannian manifold is isomorphic to a Euclidean space, a natural idea is to pick a location $\mu \in \mathcal{M}$ and define a zero-mean distribution in $\mathcal{T}_\mu\mathcal{M}$. Any mapping from $\mathcal{T}_\mu\mathcal{M}$ then induces an implicit distribution over $\mathcal{M}$, but in particular, the Exp mapping is an appealing choice due to its relationship to geodesic distances $\|v\|_\mu^2 = d^2(\mu, \mathrm{Exp}_\mu(v))$. Sampling from wrapped distributions amounts to sampling from the tangent distributions and applying the Exp mapping. Assessing probabilities is possible for manifolds where Exp is invertible and it's Jacobian tractable, *e.g.*, on the hyperbolic space, where Nagano et al. (2019) successfully used wrapped normal distributions. However, for many important manifolds, Exp is not invertible due to curvature, and many tangent points can map to the same manifold location, leading to

---

[1]We urge caution with the different types of logarithms and exponentials used throughout the paper: $\mathrm{Log}_x$, $\mathrm{Exp}_x$ refer to the mappings between manifold and tangent space. We use exp and log for the base-$e$ natural exponential and logarithm on real and complex numbers, and by extension also for the matrix exponential and logarithm. The notation $\log_\beta$, $\exp_\beta$, introduced in §2.3, are generalizations of log and exp from nonextensive statistical mechanics, and *not* base-$\beta$ operations.

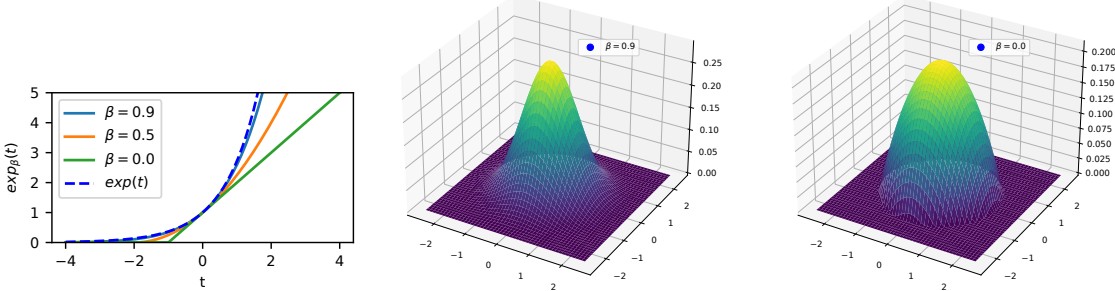

**Figure 2:** Left: the Tsallis $\exp_\beta$ recovers the usual exp as $\beta \to 1$. Right: isotropic $\beta$-Gaussian density.

generally intractable infinite summations. Constraining or renormalizing an arbitrary tangent distribution over the injectivity domain of Exp would solve this problem, at the cost of complicating both sampling and normalizing constant calculation, outside of simple isotropic cases. Instead, we show how a careful choice of compactly-supported $\beta$-Gaussian distributions in tangent space can keep the support within the injectivity domain with straightforward calculations, leading to a computationally friendly wrapped distribution that enjoys all the benefits mentioned in this section: a flexible, anisotropic, geometry-aware distribution with efficient reparametrized sampling and tractable exact probability assessment.

### 2.3 Sparse continuous distributions

To ensure that the injectivity constraint is satisfied, the tangent distribution should have bounded, controllable support. Martins et al. (2022) study such distributions and propose the $\beta$-Gaussian family, a generalization of the Gaussian (normal) distribution. We summarize their construction here and refer readers to Martins et al. (2022, §6) for more details.

**Tsallis generalized entropies.** Whereas the Gaussian is deeply connected to the Shannon-Boltzmann-Gibbs differential entropy $H[p] = \mathbb{E}_p[-\log p(t)]$, $\beta$-Gaussians are generated by the Tsallis entropies (Tsallis, 1988), from nonextensive statistical mechanics. Consider, for $\beta \neq 1$, and denoting $[\cdot]_+ := \max(0, \cdot)$,

$$\exp_\beta(t) := [1 + (1-\beta)t]_+^{\frac{1}{1-\beta}}, \qquad \log_\beta(t) := (t^{1-\beta} - 1)/(1-\beta). \tag{2}$$

Notice that $\lim_{\beta \to 1} \exp_\beta(t) = \exp(t)$ and $\lim_{\beta \to 1} \log_\beta(t) = \log(t)$, and so we extend both functions by continuity at $\beta = 1$ as the usual exp and log. The Tsallis negative entropy is defined as

$$\Omega_\beta[p] := (1/(2-\beta))\mathbb{E}_p[\log_\beta p(t)]. \tag{3}$$

**$\beta$-Gaussians.** Given a location parameter $u \in \mathbb{R}^n$ and a symmetric positive semidefinite scale parameter $\Sigma$, the multivariate Gaussian $\mathcal{N}(u, \Sigma)$ has density $p(v) = \exp(-\frac{1}{2}(v-u)^\top \Sigma^{-1}(v-u) - A(\Sigma))$, where $A(\Sigma) = \log\sqrt{(2\pi)^n|\Sigma|}$ is a normalizing constant that ensures the density integrates to 1. The multivariate $\beta$-Gaussian (Martins et al., 2022) is instead defined by the following density function:

$$v \sim \mathcal{N}_\beta(u, \Sigma) \quad \iff \quad p(v) := \exp_\beta\left(-\frac{1}{2}(v-u)^\top \Sigma^{-1}(v-u) - A_\beta(\Sigma)\right), \tag{4}$$

where $\beta$ is the Tsallis entropy parameter and the normalization constant is given by

$$A_\beta(\Sigma) := \frac{1}{\beta-1} - \frac{R^2}{2}|\Sigma|^{-(n+2/(1-\beta))^{-1}}, \quad \text{where} \quad R := \left(\frac{\Gamma\left(\frac{n}{2} + \frac{\beta}{\beta-1/2}\right)}{\Gamma\left(\frac{\beta}{\beta-1/2}\right)\pi^{\frac{n}{2}}} \cdot \left(\frac{2}{1-\beta}\right)^{\frac{1}{1-\beta}}\right)^{\frac{1-\beta}{2+(1-\beta)n}}. \tag{5}$$

In the limit of $\beta \to 1$, $\exp_\beta$ recovers exp and $\lim_{\beta \to 1} A_\beta(\Sigma) = A(\Sigma)$, and so, by continuity, the 1-Gaussian is the usual Gaussian. For $\beta = 0$, we obtain a truncated paraboloid; for $\beta \to -\infty$, the $\beta$-Gaussian converges to

the uniform distribution on an ellipsoid. Introducing a convenience parameter $\tilde{\Sigma}$, Martins et al. (2022) show that $\beta$-Gaussians are elliptical distributions satisfying

$$y \sim \mathcal{N}_\beta(u, \Sigma) \iff y = u + \tilde{\Sigma}^{\frac{1}{2}} z, \text{ where } z \sim \mathcal{N}_\beta(0, I), \ \tilde{\Sigma} := |\Sigma|^{-\frac{1}{n + \frac{2}{1-\beta}}} \Sigma, \tag{6}$$

and that the support (*i.e.*, the set of points with nonzero density) of the general $\beta$-Gaussian is an ellipsoid:

$$\text{supp}(\mathcal{N}_\beta(u, \Sigma)) = \{v : (v - u)^\top \tilde{\Sigma}^{-1}(v - u) < R^2\}, \tag{7}$$

where the radius $R$ (Eq. (5)) only depends on $\beta$ and the dimension of the space.

**Fenchel-Young losses.** For $v \notin \text{supp}(\mathcal{N}_\beta(u, \Sigma))$, $p(v) = 0$ and therefore gradient-based fitting cannot be used to fit the parameters by maximizing log-likelihood. Instead, Blondel et al. (2020) propose the framework of *Fenchel-Young (FY) losses*, derived as a natural learning objective for distributions induced by the Tsallis entropies. In particular, when $p$ is a $\beta$-Gaussian and $q$ is an arbitrary density, the FY loss between $p$ and $q$ is:

$$\ell(q : p) = \Omega_\beta[q] + \Omega_\beta^*[f] - \mathbb{E}_{q(v)}[f(v)], \tag{8}$$

where $(\cdot)^*$ denotes the Fenchel conjugate (Borwein & Lewis, 2010, §3.3), and $f(v) = -\frac{1}{2}(v - u)^\top \Sigma^{-1}(v - u)$ is the function generating the $\beta$-Gaussian (Martins et al., 2022, Definition 3). When $q$ is an empirical distribution, $\Omega_\beta[q]$ is infinite, but constant *w.r.t.* the learnable parameters of $f$, motivating the cross-FY loss:

$$\ell^\times(q : p) := \Omega_\beta^*[f] - \mathbb{E}_{q(v)}[f(v)]. \tag{9}$$

The functions $\Omega_\beta$ and $\Omega_\beta^*$ have closed-form expressions for $\beta$-Gaussians (Appendix A.1). As $\beta \to 1$, $L$ recovers the KL divergence and $\ell^\times$ recovers the cross-entropy loss. Importantly, $\ell(q : p)$ is finite when the support of $q$ and $p$ don't match, and $\ell^\times(q : p)$ is finite even when $q$ has infinite entropy outside of the support of $p$. This allows learning and gradient-based modeling.

# 3 The wrapped $\beta$-Gaussian distribution

## 3.1 Construction

We propose a tractable distribution on general Riemannian manifolds by wrapping a suitably-parametrized $\beta$-Gaussian defined in tangent space.

**Definition 1** (Wrapped $\beta$-Gaussian). *A random variable $y$ has wrapped $\beta$-Gaussian distribution with location $\mu \in \mathcal{M}$ and scale $\Sigma$, if $y = \text{Exp}_\mu(v)$, where $v \sim \mathcal{N}_\beta(0, \Sigma)$. We write $y \sim \mathcal{WN}_\beta(\mu, \Sigma)$.*

As a wrapped distribution, we have that $\text{Cov}[y] = \text{Cov}[v]$ and the Karcher mean is $\mu$ (Chevallier et al., 2022), which is further discussed in §5.1.

For general wrapped distributions, including the wrapped Gaussian ($\beta = 1$), assessing the probability density value $p(y)$ is intractable. This is due to the general non-injectivity of $\text{Exp}_\mu$, and is often sidestepped using approximations that assume small $\|\Sigma\|$. With wrapped $\beta$-Gaussians, we can have exactly tractable density estimation, alongside all other benefits of wrapped distributions, by parametrizing $\Sigma$ to ensure the support of $v$ is within the injectivity radius at $\mu$.

To derive the necessary parametrization we use the following result on the $\beta$-Gaussians:

**Lemma 1.** *Let $\tilde{\Sigma} = |\Sigma|^{-\frac{1}{n+2/(1-\beta)}} \Sigma$, and $R$ as in Eq. (26). If the maximal eigenvalue $\lambda_{\max}(\tilde{\Sigma}) < \frac{r^2}{R^2}$ then $\text{supp}\,\mathcal{N}_\beta(0, \Sigma) \subset B(0, r)$*

The proof can be found in Appendix A.2. In practice, it is often more numerically convenient to parameterize $\beta$-Gaussians directly using $\tilde{\Sigma}$, so we may use the two interchangeably.

Equipped with the result of Lemma 1, we derive the following tractable expression for densities, provided the scale is within a generous constraint set. By varying $\beta, \mu$, and $\tilde{\Sigma}$, we can obtain an expressive family with adaptive support within the injectivity domain around any point.

**Proposition 1.** *Let* $y \sim \mathcal{WN}_\beta(\mu, \Sigma)$ *be a wrapped $\beta$-Gaussian on* $\mathcal{M}$, *with* $\tilde{\Sigma}$ *as in Lemma* 1.
*If* $\lambda_{\max}(\tilde{\Sigma}) < \frac{\operatorname{inj}_\mu(\mathcal{M})^2}{R^2}$, *then the density of $y$ has the exact expression*

$$p(y) = \exp_\beta\left(-\frac{1}{2}\operatorname{Log}_\mu(y)^\top \Sigma^{-1} \operatorname{Log}_\mu(y) - A_\beta(\Sigma)\right)\left|\frac{\partial \operatorname{Log}_\mu(y)}{\partial y}\right|. \tag{10}$$

*Proof.* From Lemma 1 with $r = \operatorname{inj}_\mu(\mathcal{M})$, we have that the support of $\mathcal{N}_\beta(0, \Sigma)$ lies strictly in the injectivity domain of $\mathcal{M}$ at $\mu$, on which $\operatorname{Exp}_\mu$ and $\operatorname{Log}_\mu$ are bijections. The change-of-density formula (push-forward) yields the desired result. $\square$

Like $\beta$-Gaussians, wrapped $\beta$-Gaussians enjoy efficient tractable reparametrized sampling, since the Exp mapping is differentiable. This means all necessary building blocks for deep generative models are available.

### 3.2 Fenchel-Young losses

Due to the compact support of $\beta$-Gaussians, $p(y)$ is zero on points outside of the support. For fitting the distribution parameters to data, therefore, the usual gradient-based *maximum likelihood* approach does not apply. In Euclidean space, Fenchel-Young losses (§2.3) address this concern. In this section, we study their extension to wrapped distributions.

**Losses in tangent space.** On $\mathcal{T}_\mu\mathcal{M}$, the natural learning objective for a $\beta$-Gaussian is the cross-Fenchel-Young loss, which for a single target point $v \in \mathcal{T}_\mu\mathcal{M}$ takes the value

$$\ell^\times(\delta_v : p) = (1 - \beta)\Omega_\beta[p] + A_\beta(\Sigma) + \frac{1}{2}v^\top \Sigma^{-1} v. \tag{11}$$

The derivation is provided by Martins et al. (2022, Proposition 18). For $x \in \mathcal{M}$, we can therefore directly define the tangent loss

$$\ell_t^\times(x) := \ell^\times(\delta_{\operatorname{Log}_\mu(x)} : p) = (1 - \beta)\Omega_\beta[p] + A_\beta(\Sigma) + \frac{1}{2}\operatorname{Log}_\mu(x)^\top \Sigma^{-1} \operatorname{Log}_\mu(x). \tag{12}$$

**Wrapped Fenchel-Young losses.** While the tangent space loss is adequate and we find it to perform well in practice, it does not account for the distortion induced by wrapping. Notably, while in tangent space $\lim_{\beta \to 1} L^\times(\delta_v, p) = -\log p(v)$, the wrapped case $\lim_{\beta \to 1} \ell_t(x) \neq -\log p(x)$, as the Jacobian term in Eq. (10) is not accounted for. The correct way to incorporate a Jacobian term $J(x)$ from a change of density, under the Tsallis non-extensive system, is not straightforward from Eq. (11). We next show how to relate the Fenchel-Young losses with the probability measure directly by proposing a new rearrangement.

Notice that, in tangent space, if $p(v) > 0$ then

$$\ell^\times(\delta_v : p) = (1 - \beta)\Omega_\beta[p] - \log_\beta p(v), \tag{13}$$

where $p(v) = \exp_\beta\left(f(v) - A_\beta(\Sigma)\right) = \exp_\beta(h(v))$. Substituting $p(v) = p(\operatorname{Log}(x)) \cdot |J(x)|$ we can apply

$$\log_\beta(ab) = \log_\beta(a) + \log_\beta(b) + (1 - \beta)\log_\beta(a)\log_\beta(b) = b^{1-\beta}\log_\beta(a) + \log_\beta(b). \tag{14}$$

Then, introducing $h(v) := -1/2\, v^\top \Sigma^{-1} v - A_\beta(\Sigma)$, we get the following the wrapped loss:

$$L^\times(x) := (1 - \beta)\Omega_\beta[p] - |J(x)|^{1-\beta}h(\operatorname{Log}_\mu(x)) - \log_\beta(|J(x)|). \tag{15}$$

Contrasting Eq. (15) with $\ell_t^\times(x) = (1 - \beta)\Omega_\beta[p] - h(\operatorname{Log}_\mu(x))$ reveals that the Jacobian has both multiplicative and additive contributions; moreover, the loss satisfies the desired property $\lim_{\beta \to 1} L^\times(x) = -\log p(x)$.

Outside of the support, $\log_\beta$ and $\exp_\beta$ are not inverses of each other, but we can write $L^\times(\delta_v : p) = (1 - \beta)\Omega_\beta[p] - \log_\beta p_0(v)$ where $p_0(v) = [1 + (1 - \beta)h(v)]^{1/(1-\beta)}$ is the "unclipped" version of $p$, continuous and agreeing with $p(v)$ on the support. Since $\log_\beta p_0(v) = h(v)$, we may use Eq. (15) on all of $\mathcal{M}$.

We stress that even outside of the compact support of the wrapped $\beta$-Gaussians, the proposed FY losses are finite and effective in learning on the entire domain of $\text{Log}_\mu(x)$ (*e.g.*, for spheres, this excludes only the antipode $-\mu$).

### 3.3 Parametrizations of the scale parameter

Instantiating (and modeling with) a wrapped $\beta$-Gaussian $\mathcal{WN}_\beta(\mu, \Sigma)$ requires the specification of a $\beta$-Gaussian $\mathcal{N}_\beta(0, \Sigma)$ such that (i) its support of $\mathcal{N}_\beta(0, \Sigma)$ is on the tangent space $\mathcal{T}_\mu\mathcal{M}$ and (ii) the spectrum of its scale is bounded following Proposition 1. In this section, we detail several ways to achieve both desiderata efficiently.

To ensure that a tangent space $\beta$-Gaussian distribution remains within the injectivity radius, we parametrize the eigenvalues of $\tilde{\Sigma}$ to have values between 0 and $\lambda_{\max} = \frac{\text{inj}(\mathcal{M})^2}{R^2}$ from Proposition 1, using the sigmoid function in log-domain: we fit a real parameter $s \in \mathbb{R}^m$ and set

$$\log \tilde{\lambda}_j = \log \lambda_{\max} - \log(1 + \exp(-s_j)). \tag{16}$$

We may then use $\tilde{\Sigma} = \text{diag}(\exp \tilde{\lambda})$ as a feasible diagonal scale matrix. While we do not report experiments with full-scale matrices, the same strategy may be used, introducing an additional orthogonal parameter matrix for the eigenvectors.

**Local coordinates.** The scale parameter can be defined as a $n$-dimensional matrix in some chosen basis of $\mathcal{T}_\mu\mathcal{M}$. This requires an arbitrary choice of basis which should depend on $\mu$ continuously, leads to complicated gradients, and makes the comparison between distributions at different points challenging. We propose identifying a point $p \in \mathcal{M}$ (a pole) and parametrizing a matrix $\Sigma$ (or $\tilde{\Sigma}$) in $\mathcal{T}_p\mathcal{M}$. Then we may implicitly define $\Sigma_\mu$ via parallel transport map such that

$$PT_{p \to \mu} v \sim \mathcal{N}_\beta(0, \Sigma_\mu) \text{ where } v \sim \mathcal{N}_\beta(0, \Sigma). \tag{17}$$

For a Riemannian manifold $\mathcal{M}$ embedded in $\mathbb{R}^n$, parallel transport is an isometry, therefore the two densities are equivalent and we can assess the probability of any vector in $\mathcal{T}_\mu$ by first applying $PT_{\mu \to p}$. In other words, in embedded coordinates $PT_{p \to \mu}$ has an orthogonal matrix representation $Q_\mu$, and $|\Sigma_\mu| = |Q_\mu \Sigma Q_\mu^{-1}| = |\Sigma|$.

**Ambient coordinates.** Alternatively, we may specify $\Sigma$ as an $n$-by-$n$ matrix in ambient space and denote $P_\mu$ the rank-$d$ orthogonal projection matrix onto $\mathcal{T}_\mu\mathcal{M}$, and set $\Sigma_\mu = P_\mu \Sigma P_\mu$. This removes the dependency on an arbitrary pole and leads to a more interpretable parametrization. In general, carrying out calculations with $P_\mu \Sigma P_\mu$ requires a costly factorization even when $\Sigma$ is already decomposed or diagonal. In §4.2 we show efficient expressions that avoid costly factorization when working with the sphere manifold $\mathbb{S}^n$. Since projection matrices have eigenvalues 0 and 1, it is sufficient to constrain the ambient $\Sigma$ as described above, to obtain a feasible scale matrix after projection.

## 4 Instantiations on specific manifolds

### 4.1 Geometric toolbox

Table 1 summarize all the necessary geometric tools to instantiate the wrapped $\beta$-Gaussian on two manifolds: the hypersphere ($\mathbb{S}^n$) and the manifold of rotations, also known as the special orthogonal group ($SO(n)$).

### 4.2 Hypersphere $\mathbb{S}^{n-1}$ embedded in $\mathbb{R}^n$

**Relative coordinates.** Without loss of generality, we choose the north pole $p = (0, \dots, 0, 1)$ as a reference. The corresponding tangent space is $\mathcal{T}_p\mathcal{M} = \{(v, 0) | v \in \mathbb{R}^{n-1}\}$ where we can easily parametrize a scale parameter by removing the unused last coordinate. We rotate the space via parallel transport, and implement it as a composition of two Householder reflections in $\mathbb{R}^n$: $PT_{p \to \mu} = R_\mu \circ R_{\frac{\mu+p}{\|\mu+p\|}}$ (Algorithm 1). $R_{\frac{\mu+p}{\|\mu+p\|}}$ maps a point from $\mathcal{T}_p\mathbb{S}^{n-1}$ to $\mathcal{T}_\mu\mathbb{S}^{n-1}$, and $R_\mu$ maps any point on $\mathcal{T}_\mu\mathbb{S}^{n-1}$ to itself, so can be considered as identity.

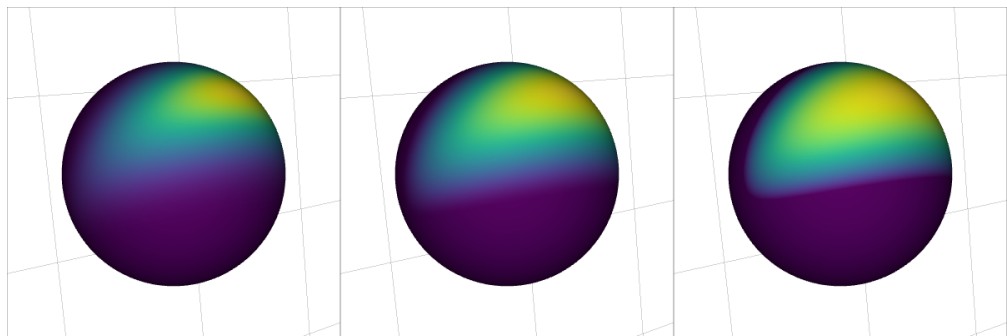

**Figure 3:** Wrapped $\beta$-Gaussian density on $\mathbb{S}^2$ for $\beta = 0.9$, $\beta = 0.5$, $\beta = 0$ (from left to right).

**Table 1:** Geometric toolbox for the sphere and rotation manifolds. For the Jacobian on $SO(n)$, $\pm i\theta_j$ are the eigenvalues of $X^\top V$, $m = \lfloor n/2 \rfloor$, and $\gamma = n \mod 2$.

|  | Hypersphere $\mathbb{S}^n$ (Appendix B) | Special orthogonal group $SO(n)$ (Appendix D) |
|---|---|---|
| Tangent space | $\mathcal{T}_x \mathbb{S}^n = \{v \in \mathbb{R}^n, \langle x, v \rangle = 0\}$ | $\mathcal{T}_X SO(n) = \{V \in \mathbb{R}^{n \times n} : V^\top X + X^\top V = 0\}$ |
| Injectivity radius | $\operatorname{inj} \mathbb{S}^n = \pi$ | $\operatorname{inj} SO(n) = \pi\sqrt{2}$ |
| Exponent | $\operatorname{Exp}_x(v) = \cos(\|v\|_2) x + \sin(\|v\|) \frac{v}{\|v\|}$ | $\operatorname{Exp}_X(V) = X \exp(X^\top V)$ |
| Logarithm | $\operatorname{Log}_x(y) = \frac{\arccos(\langle y, x \rangle)}{\sqrt{1 - \langle y, x \rangle^2}} (y - \langle y, x \rangle x)$ | $\operatorname{Log}_X(Y) = X \log(X^\top Y)$ |
| Jacobian | $\left\| \frac{\partial \operatorname{Exp}_x(v)}{\partial v} \right\| = \left( \operatorname{sinc}(\|v\|_2) \right)^{n-1}$ | $\left\| \frac{\partial \operatorname{Exp}_X V}{\partial V} \right\| = \prod_{j=\gamma}^m \prod_{k=1}^m \operatorname{sinc}^2 \left( \frac{\theta_j - \theta_k}{2} \right) \operatorname{sinc}^2 \left( \frac{\theta_j + \theta_k}{2} \right)$ |

**Ambient parametrization.** On the sphere, we provide a result that allows the ambient space parametrization can be used efficiently. We may posit a scale matrix $\tilde{\Sigma} \in \mathbb{R}^{n \times n}$ and observe that the projection operator $P_\mu = I - \mu\mu^\top$ is a rank-one matrix. The projected scale $\tilde{\Sigma}_\mu = P_\mu \tilde{\Sigma} P_\mu$ has the subspace $\mathbb{R}\mu$ in its null space, and we have $\lambda_{\max}(\Sigma_\mu) \le \lambda_{\max}(\Sigma)$. Thanks to its structure, we can derive exact expressions for the pseudoinverse $\tilde{\Sigma}_\mu^+$ and the pseudodeterminant $|\tilde{\Sigma}_\mu|_+$, in terms of the ones of $\tilde{\Sigma}$.

**Proposition 2.** *Let $S$ be a $n$-by-$n$ positive semidefinite matrix parameter and $P = (I - xx^\top)$ be the projection operator onto the hyperplane orthogonal to a unit vector $x$. Let $R = I - S^+(S^+)^\top$ be the projection onto the kernel of $S$, and $\beta = x^\top R x$. Then, we have:*

$$(i) \qquad |PSP|_+ = \begin{cases} |S| \cdot x^\top S^+ x, & \beta = 0, \\ |S| \cdot \beta, & \beta \ne 0. \end{cases}$$

$$(ii) \qquad (PSP)^+ = \begin{cases} S^+ - \frac{S^+ xx^\top S^+}{x^\top S^+ x}, & \beta = 0, \\ \left( I - \frac{1}{\beta} Rxx^\top \right) S^+ \left( I - \frac{1}{\beta} Rxx^\top \right)^\top, & \beta \ne 0. \end{cases}$$

The proof, and further implementation details, can be found in Appendix C. If $\tilde{\Sigma}$ is diagonal, this result allows us to perform all calculations needed for assessing densities or FY losses in $O(n)$, and if $\tilde{\Sigma}$ is stored in a factorized form, in $O(n^2)$. Without this result, a cubic-cost eigendecomposition of $\tilde{\Sigma}_\mu$ would be required. If $\tilde{\Sigma}$ is guaranteed full-rank (*e.g.*, by parametrization), then only the $\beta = 0$ case is needed.

### 4.3 The Special Orthogonal manifold of rotation matrices $SO(n)$

The manifold $SO(n)$ is a well-studied Lie group. As an embedded submanifold of $\mathbb{R}^{n \times n}$, its elements are identified with orthogonal matrices with determinant 1, *i.e.*, rotation matrices. This makes it valuable in practical applications involving rotations and alignment between spaces. The tangent space at the identity matrix is the vector space of skew-symmetric matrices, with dimensionality $n(n-1)/2$. On this manifold, we only employ a pole-based parametrization. The Exp mapping involves the matrix exponential, and its Jacobian has an efficient expression which we present, along with the rest of the toolbox, in Appendix D.

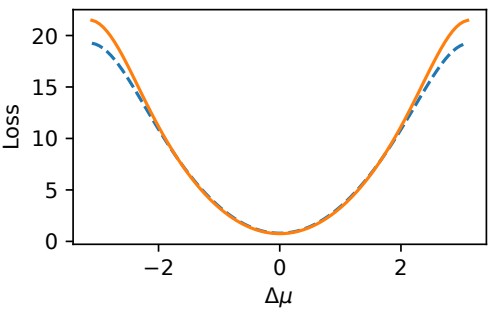 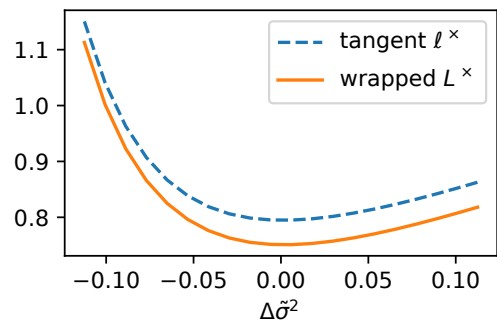

**(a)** Fenchel-Young losses *w.r.t.* shifting the true location $\mu$: $\mu' = \mathrm{Log}_\mu(PT_{p\to\mu}(\Delta\mu, 0))$.

**(b)** Fenchel-Young losses *w.r.t.* shifting the true $\tilde{\sigma}^2$ of the $\mathcal{WN}_\beta(\mu, \tilde{\sigma}^2 I)$.

**Figure 4:** Visualization of wrapped $L^\times$ and tangent $\ell^\times$ losses *w.r.t.* shifts of (a) location, and (b) $\tilde{\sigma}^2$ parameter of $\mathcal{WN}_\beta(\mu, \tilde{\sigma}^2 I)$. $\mu$ is random on the $\mathbb{S}^2$, and the trace of $\mathrm{Cov}_\mu = 0.3$. Losses are averaged over 10000 samples from the true $\mathcal{WN}_\beta$ distribution ($\beta = 0.9$). Additionally, in Appendix B.4, we provide the plot for $\mathbb{S}^{29}$, where we observe the higher influence of the Jacobian correction term from Eq. (15).

## 5 Experiments

We conduct a series of experiments to showcase learning with wrapped $\beta$-Gaussians.

In our experiments, we use $\beta = 0.9$ as a default value, for which the distribution shape is relatively close to a Gaussian distribution ($\beta = 1.0$), but with finite support size.

### 5.1 Parameter estimation: moment matching and FY loss minimization

As Chevallier & Guigui (2020) note, one of the important advantages of wrapped distributions is a more straightforward relationship between the moments and the parameters of the distribution.

For Riemannian manifolds, the Euclidean definition of means does not apply, since manifold points cannot be summed. A natural definition instead comes from a center of mass optimization problem:

$$\mu = \mathrm{Mean}[x] = \arg\min_{y \in \mathcal{M}} \mathbb{E}_x[d(y, x)^2] \approx \arg\min_{y \in \mathcal{M}} \frac{1}{N} \sum_{i=1}^{N} d(y, x_i)^2. \tag{18}$$

Over $\mathbb{R}^n$ this recovers the expression $\mu = \mathbb{E}_x[x]$. For a general Riemannian manifold $\mathcal{M}$ the solution may not be unique. A global minimizer of Eq. (18) is known as a Fréchet mean, while a local minimizer is known as a Karcher mean.

The covariance is defined in tangent space and thus matches the one of the tangent distributions (Chevallier et al., 2022, eq. 2.6):

$$\mathrm{Cov}_\mu[x] = \mathbb{E}_x[\mathrm{Log}_\mu(x) \mathrm{Log}_\mu(x)^\top] \approx \frac{1}{N} \sum_{i=1}^{N} \mathrm{Log}_\mu(x_i) \mathrm{Log}_\mu(x_i)^\top. \tag{19}$$

In Euclidean space, the asymptotics of estimators from empirical measurements is well studied; on manifolds the situation is more difficult. Some central limit theorems for Frechét means (Bhattacharya & Patrangenaru, 2005; Pennec, 2019) apply under the Karcher-Kendall concentration conditions (Karcher, 1977; Kendall, 1989), which imply existence and uniqueness of the Frechét mean when the support is strictly contained inside a geodesic ball $B(\mu, \frac{1}{2}\min(\mathrm{inj}_\mathcal{M}, \pi/\sqrt{\kappa}))$, where $\kappa$ is an upper bound of the sectional curvature. Wrapped $\beta$-Gaussians can be readily constrained to satisfy this condition.

Although we can use the moment matching method described above to directly search for a Karcher mean[2] and estimate empirical covariance using Eq. (19), applying gradient optimization to Fenchel-Young losses is

---

[2]We use the *geomstats* package (Miolane et al., 2020) to find Karcher means on the sphere.

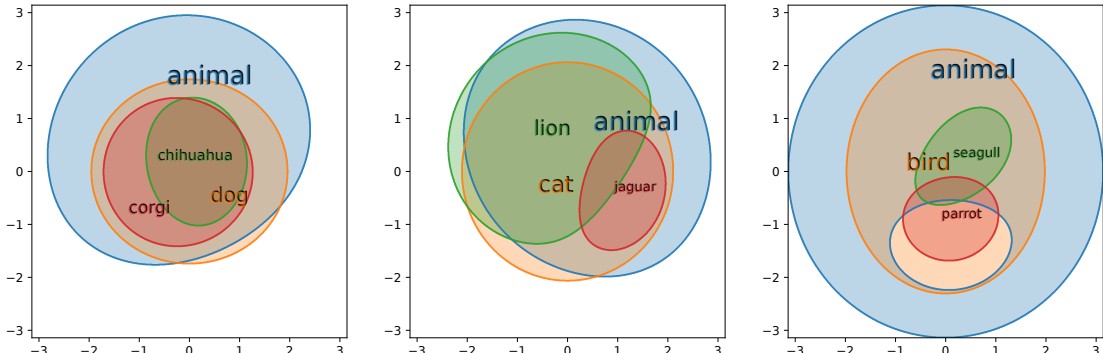

**Figure 6:** Support of the embeddings trained on lexical entailment task: pole parametrization. For the visibility we split the plot into three subplots according to the subtrees of nodes *dog*, *cat*, and *bird* (from left to right). The supports are sent to the tangent space of the words *dog*, *cat*, *bird* via logarithm map.

an alternative which also works as part of end-to-end deep learning modeling. We visualize the Fenchel-Young loss in the Figure 4. In Appendix B.4 we demonstrate fitting a $\beta$-Gaussian to samples, and show that indeed minimizing FY losses converges to the true distribution.

## 5.2 Modeling hierarchies on the sphere with FY losses

In this experiment, we demonstrate Fenchel-Young losses for modeling hierarchical relationships between distributional embeddings. We employ a synthetic hierarchy inspired by lexical entailment (Figure 5) and embed each word as a wrapped $\beta$-Gaussian. Vilnis & McCallum (2015) learn distributional embeddings with KL divergences between $\mathbb{R}^n$ Gaussians. Inspired by Muzellec & Cuturi (2018), we consider compact-support elliptical distributions, so that we may visualize the embedding of a word as the support of the distribution. Motivated by the relationship between the FY loss and the KL divergence, we assign a $\mathcal{WN}_\beta$ distribution $p_w$ for each word $w$, and minimize:

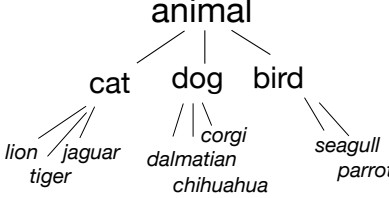

**Figure 5:** Synthetic dataset emulating lexical entailment.

$$L(w) \coloneqq L(p_w : p_{\pi(w)}) - \gamma \log \sum_{w'} \exp L(p_w : p_{w'}), \tag{20}$$

where the term $L(p_w : p_{\pi(w)})$ is calculated for words $w$ and its parent $\pi(w)$, and $L(p_w : p_{w'})$ is a loss for a negative pair (no entailment edge), where $w'$ is a negative sample, that is not an ancestor of w. The $k = 3$ negatives are sampled with a replacement for each $w$ in batch, and $L(w)$ is averaged over the batch. We train for 30 iterations with a learning rate of 0.05 and use full-batch training. We use $\gamma = 0.1$ to prevent $L(p_w : p_{w'})$ from dominating the learning objective when positive $L(w : p)$ becomes close to zero. The deep connection between $\beta$-Gaussians and FY losses allows us to learn with finite gradients even when supports are disjoint, unlike the KL divergence; we can then more directly generalize the approach of Vilnis & McCallum (2015) without switching to the Wasserstein geometry like Muzellec & Cuturi (2018).

**Results.** In Figure 6, we visualize the learned probabilistic embeddings. We observe, that the learned embeddings tend to organize the hierarchical structure: support of the superordinate incorporating the support of hyponym.

## 5.3 Anisotropic hyperspherical VAE on MNIST

In this experiment, we evaluate the wrapped $\beta$-Gaussian on dynamically binarized MNIST (Salakhutdinov & Murray, 2008) reconstruction task following Davidson et al. (2018). We train a variational auto-encoder

(Kingma & Welling, 2014, VAE;), optimizing the evidence lower bound (ELBO) objective:

$$\log p(x) \geq \mathbb{E}_{q_\theta(z|x)}[\log p_\phi(x|z)] - \text{KL}[q_\theta(z|x) : p(z)]. \tag{21}$$

We explore modeling the latent vectors on the $n$-dimensional manifold $\mathbb{S}^n$. We define the encoder output $q_\theta(z|x)$ as wrapped $\beta$-Gaussian, and use a uniform spherical prior $p(z) \propto 1$. Even if the proposal has compact support, as long as its support is included in the support of the prior, the ELBO in Eq. (21) is finite and well-defined, therefore we may use standard probability theory and do not need to resort to Fenchel-Young losses, for which a corresponding nonextensive ELBO would be nontrivial.

We estimate the gradients of the ELBO using Monte Carlo methods. For the reconstruction part of the ELBO (the first term), we apply the reparametrization trick (Kingma & Welling, 2014) to $\mathcal{WN}_\beta$, as we can compose Eq. (6) with the differentiable Exp mapping. We estimate the KL term with the *common random numbers* strategy (Blundell et al., 2015; Owen, 2013), reusing the same sample as in the first term.

We follow the hyperparameter setting of Davidson et al. (2018) using MLP with 2 hidden layers for both the encoder and the decoder: $[256, 128]$ hidden units for the encoder and $[128, 256]$ hidden units for the decoder. We trained for 1000 epochs using the Adam optimizer (Kingma & Ba, 2015) with mini-batches of size 64, and with a linear warm-up for 100 epochs and maximum learning rate of 0.001.

**Results.** The results in Table 2 indicate that wrapped $\beta$-Gaussian VAEs perform slightly worse than the isotropic S-VAE baseline in higher dimensions $n \geq 20$, but can outperform it for smaller latent dimensions $n = 2, 5, 10$. We observe that the *embedded* parametrization has overall lower variance compared to the *relative* to the pole parametrization and scales better with dimension.

**Table 2:** Summary of the results for MNIST experiment for different latent space dimensions $\mathbb{S}^n$. We report the averaged log-likelihoods estimated via importance sampling with 500 samples (Burda et al., 2016), as well as the standard deviations. Results are averaged over 10 runs.

| $z$ dim | S-VAE | relative $\mathcal{WN}_\beta$-VAE | embedded $\mathcal{WN}_\beta$-VAE |
|---|---|---|---|
| 2 | $-132.50_{\pm 0.73}$ | $-130.10_{\pm 1.65}$ | $-131.36_{\pm 0.81}$ |
| 5 | $-108.43_{\pm 0.09}$ | $-107.18_{\pm 0.16}$ | $-107.16_{\pm 0.25}$ |
| 10 | $-93.16_{\pm 0.31}$ | $-92.60_{\pm 0.51}$ | $-92.78_{\pm 0.25}$ |
| 20 | $-89.02_{\pm 0.31}$ | $-97.94_{\pm 5.84}$ | $-90.58_{\pm 0.38}$ |
| 40 | $-90.87_{\pm 0.34}$ | $-100.49_{\pm 4.53}$ | $-95.62_{\pm 0.36}$ |

### 5.4 Multilingual embedding alignment via Bayesian orthogonal Procrustes

For most natural language processing tasks it is substantially easier to obtain monolingual data rather than cross-lingually aligned data. Word embeddings (Turian et al., 2010) provide the opportunity for finding alignments between the implicit Euclidean spaces occupied by two languages. It is common to restrict the search to linear alignments (Mikolov et al., 2013a; Dinu et al., 2015; Lazaridou et al., 2015): given a dataset of paired embeddings $(u_i, v_i) \in \mathcal{D}$, obtained using a bilingual dictionary, we seek

$$\arg\min\{\sum_i \|u_i - Xv_i\|_2^2/2 : X \in \mathcal{M} \subseteq \mathbb{R}^{n \times n}\}. \tag{22}$$

A common choice of constraint is $\mathcal{M} = SO(n)$, which restricts the space of alignments to *rotations*. As rotations preserve angles, this orthogonal constraint leads to good performance in a number of NLP tasks (Xing et al., 2015; Artetxe et al., 2016; Hamilton et al., 2016). Equation (22) is known as the *orthogonal Procrustes problem*, and a direct solution is available from the SVD of the matrix $\sum_i v_i u_i^\top$ (Schönemann, 1966). However, in some situations that require more advanced analysis, a Bayesian treatment can be more informative than point estimation. For probabilistic treatment, note that Eq. (22) is a constrained regression with unit-variance Gaussian observations:

$$p(u \mid X, v) \propto \exp\left(\|u - Xv\|^2/2\right). \tag{23}$$

We treat $X$ as a random variable over $SO(n)$, with a known prior distribution $p(X)$: for simplicity, we pick the uniform distribution under the Haar measure. Under this measure, the volume of $SO(n)$ is the product

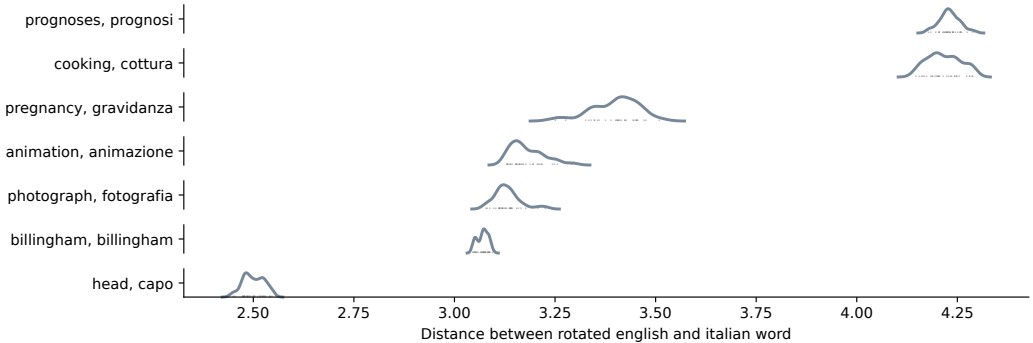

**Figure 7:** Euclidean distances between Italian embedding and rotated English embedding. English embeddings are rotated by 30 samples from the $q(X)$. The smooth distributions are 1d Gaussian kernel density estimation plots (original samples are present as well).

of the sphere surface areas for dimensions ranging from 1 to $n$, leading to: (Zhang, 2015, Theorem 2.24)[3]

$$p(X) = \frac{1}{\text{Vol}(SO(n))} = \frac{\prod_{k=1}^{n} \Gamma(k/2)}{2^{n-1}\pi^{n(n+1)/4}}. \tag{24}$$

Under the uniform prior, the *maximum a posteriori* point estimate is the same as the standard orthogonal Procrustes solution to Eq. (22), but the full posterior distribution $p(X \mid \mathcal{D})$ is intractable. We perform variational inference with a wrapped $\beta$-Gaussian approximate posterior

$$q_\theta(X) \sim \mathcal{WN}_\beta(\mu_X, \Sigma_X),$$

where $\mu_X \in SO(n)$ is a trained location parameter parametrized as $\mu_X = \exp(L - L^\top), L \in \mathbb{R}^{n \times n}$, and $\Sigma_X$ is a covariance operator which is diagonal in our chosen basis of $\mathcal{T}_I SO(n)$ (per Proposition 5), and thus parametrized as a log-domain $n(n-1)/2$-dimensional vector. We then maximize an evidence lower-bound:

$$\log p(u \mid v) \geq \mathbb{E}_{q_\theta(X)}[\log p(u \mid X, v)] - \text{KL}[q_\theta(X) : p(X)], \tag{25}$$

using reparametrized samples from $q_\theta(X)$ and the tractable expression of the density $\log q_\theta(X)$, like in §5.3.

We fit this probabilistic orthogonal Procrustes model on the English-Italian alignment dataset from Dinu et al. (2015), where the source and target spaces are independently trained continuous bag-of-words representations (Mikolov et al., 2013b) of dimensionality $n = 300$. (The dimension of $SO(n)$ is then $44\,850$.) Despite the very high dimension, we encounter no numerical issue in optimizing, starting with a random initialization, and converge to a probabilistic with precision@1 test set performance just slightly worse than the point estimate ($42.3 \pm 0.3\%$ vs. $43.3\%$). Figure 7 demonstrates the kind of analysis of some individual word translation made possible by the probabilistic treatment: some word pairs with more ambiguity (*e.g.*, *pregnancy–gravidanza*) get mapped to a wider range of values, as measured by the distance from the expected translation in the dictionary, compared to other words, like the toponym *billingham*, which is much less ambiguous. In addition to the analysis, the probabilistic treatment also allows to incorporate further domain-specific knowledge through the prior $p(X)$, which need not be uniform.

## 6   Related work

**Prescribed distributions on spheres.**   The sphere manifold $\mathbb{S}^n$ has in particular attracted much attention from the field of directional statistics (Mardia & Jupp, 1999). Possibly the most widely used distribution on $\mathbb{S}^n$ is the Langevin distribution, obtained by conditioning an isotropic Gaussian. De Cao & Aziz (2020) propose the Power Spherical distribution, which is similarly shaped to Langevin but more numerically stable in high dimensions. For anisotropic covariances, the counterpart to Langevin is the Fisher-Bingham distribution

---

[3]The theorem gives the volume of $O(n)$, which consists of two disconnected subsets, each isomorphic to $SO(n)$.

(Kent, 1982), which in general is not tractable, requiring numerical integration (Chen & Tanaka, 2021). The Spherical Normal (Hauberg, 2018) is an intrinsical Riemannian construction supporting full covariances, but neither densities nor reparametrized samples are available in general dimensions. The wrapped $\beta$-Gaussian supports full covariances without sacrificing exact and efficient density assessment and sampling.

**Wrapped distributions.** Chevallier & Guigui (2020) study wrapped distributions within full generality, and Chevallier et al. (2022) provide a general form for the required Jacobian in symmetric spaces (including spheres and $SO(n)$.) If the injectivity radius is infinite (*e.g.*, for hyperbolic space), wrapping a Gaussian leads to tractable and well-performing models in large-scale experiments (Nagano et al., 2019). For finite injectivity radii, however, the infinite support of the Gaussian leads to intractable integrals, which require either ignoring the tails (an approximation only valid for high concentrations), or truncating the distribution, which is generally not tractable (Galaz-Garcia et al., 2022). Our construction, based on $\beta$-Gaussians, leads to a practical solution for exact wrapping on manifolds with finite injectivity radii.

**Implicit distributions on manifolds** For statistical inference or modeling, more flexibility is required than even anisotropic distributions can provide. Approaches based on neural networks have been extended to Riemannian geometry, including normalizing flows (Rezende et al., 2020; Mathieu & Nickel, 2020), score-based methods (Bortoli et al., 2022), and diffusion methods (Huang et al., 2022; Okhotin et al., 2023). Sampling from general energies on manifolds is achievable by methods like Geodesic Monte Carlo (Byrne & Girolami, 2013) and Riemannian Stein Variational Gradient Descent (Liu & Zhu, 2018) (Liu & Zhu (2022) provide a review). The flexibility of these methods comes at a higher computational cost.

## 7  Conclusion

We introduce wrapped $\beta$-Gaussians: a flexible family of distributions on Riemannian manifolds, with efficient expressions for sampling, learning, and exact assessment of density. We adapt the Fenchel-Young losses, the natural learning criterion for $\beta$-Gaussians, to take into account the manifold curvature. We instantiate our construction on spheres and rotations, deriving new expressions for an ambient-space parametrization on spheres, and experimentally validate the utility of $\beta$-Gaussians for learning, variational auto-encoding, and high-dimensional variational inference for orthogonal Procrustes. We release an open-source implementation that we hope will aid researchers and practitioners exploring new applications and other manifolds.

## 8  Limitations

In manifold statistics, there is a classical trade-off between the wrapped and intrinsic approaches. Intrinsic distributions have energies defined directly by geodesic distances and have more intuitive contours and optimization behavior when far from the location. Wrapped distributions incur a distortion from tangent space, but in exchange are algorithmically friendlier. The advantage of $\beta$-Gaussian is that by restricting the support size we may keep all probability mass in regions of low distortion, if desired.

Our work is directly applicable to a limited but fairly wide choice of Riemannian manifolds, where standard tools for wrapped distribution are available: Exp, Log maps, determinant of their Jacobian, and an expression for the injectivity radius. This includes a widely used class of symmetric Riemannian manifolds (Chevallier et al., 2022), of which $\mathbb{S}^n$, $SE(n)$, $SO(n)$ are particular examples.

By construction, the support size of the wrapped $\beta$-Gaussian is bounded by the injectivity radius at a particular point on the manifold. Depending on the injectivity radius, there may be an empirical distribution that cannot be fit by a single wrapped $\beta$-Gaussian. In such a situation, mixture distributions may a be promising extension of our approach.

## Acknowledgements

This publication is part of the project VI.Veni.212.228 of the research programme 'Veni', which is financed by the Dutch Research Council (NWO); and is part of 'Hybrid Intelligence: augmenting human intellect'

(https://hybrid-intelligence-centre.nl) with project number 024.004.022 of the research programme 'Gravitation' which is (partly) financed by the Dutch Research Council (NWO). We thank the action editor and the anonymous reviewers for their helpful comments and discussion, which have improved this text. We are thankful to Wilker Aziz, Caio Corro, Mario Figueiredo, André Martins, Dmitry Molchanov, and all members of the Language Technology Lab for helpful feedback for their valuable feedback on earlier drafts of this work. This work is built on open-source software; we acknowledge the scientific Python stack (Van Rossum & Drake, 2009; Oliphant, 2006; Walt et al., 2011; Virtanen et al., 2020; Paszke et al., 2019).

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

# Appendix

## A Sparse continuous distributions

### A.1 Expressions

For completeness, we cite here some expressions given by Martins et al. (2022).

The radius of a standard $\beta$-Gaussian is:

$$R = \left( \frac{\Gamma\left(\frac{n}{2} + \frac{\beta}{\beta - 1/2}\right)}{\Gamma\left(\frac{\beta}{\beta - 1/2}\right) \pi^{\frac{n}{2}}} \cdot \left(\frac{2}{1 - \beta}\right)^{\frac{1}{1-\beta}} \right)^{\frac{1-\beta}{2+(1-\beta)n}}. \tag{26}$$

The normalizing constant of a $\beta$−Gaussian has expression

$$A_\beta(\Sigma) = \frac{1}{\beta - 1} - \frac{R^2}{2} |\Sigma|^{-\frac{1}{n+2/(1-\beta)}}. \tag{27}$$

The Tsallis negentropy of a $\beta$-Gaussian $p$ is

$$\Omega_\beta[p] = -\frac{1}{(2 - \beta)(1 - \beta)} + \frac{R^2 |\Sigma|^{-\frac{1}{n+2/(1-\beta)}}}{2(2 - \beta) + n(1 - \beta)}. \tag{28}$$

If $f$ is the function that induces the sparse distribution $p$, then we have the relationship (Martins et al., 2022, Proposition 10)

$$\Omega_\beta^*[f] = (1 - \beta)\Omega_\beta[p] + A_\beta(\Sigma). \tag{29}$$

In all expressions, if $\Sigma$ is rank-deficient, we may use the pseudodeterminant in order to obtain a distribution in a lower-dimensional subspace with respect to the appropriate Lebesgue measure (Gelbrich, 1990)

### A.2 Eigenvalues and support: Proof of Lemma 1

When the scale of $\beta$-Gaussian is parametrized by $\tilde{\Sigma}$ it is easy to control for the support size of $\beta$-Gaussian to be strictly less than a desired radius $r$. Let $\tilde{\Sigma} = P\Lambda P^\top$, for $PP^\top = I$, then:

$$\sup_{t^\top \tilde{\Sigma}^{-1} t < R^2} \|t\|_2^2 = \sup_{t^\top \Lambda^{-1} t < R^2} \left\|P^\top t\right\|_2^2 = \sup_{\|t\|_2^2 < R^2 \lambda_{\max}(\tilde{\Sigma})} \|t\|_2^2 = R^2 \lambda_{\max}(\tilde{\Sigma}) < r^2.$$

## B The wrapped $\beta$-Gaussian on the sphere

### B.1 Inverse exponential map

In this section, we derive the inverse of the exponential map,

$$y = \text{Exp}_x(v) = \cos(\|v\|_2)x + \sin(\|v\|_2)\frac{v}{\|v\|_2}.$$

We seek $v \in \text{Exp}_x^{-1}(y)$. The first step is to identify the possible values for $\|v\|_2$.

$$\langle y, x \rangle = \cos(\|v\|_2) \underbrace{\langle x, x \rangle}_{=1} + \sin(\|v\|_2)\frac{1}{\|v\|_2} \underbrace{\langle v, x \rangle}_{=0} \tag{30}$$

$$\|v\|_2 \in \{\arccos\langle y, x \rangle + 2k\pi \mid k \in \mathbb{Z}\}.$$

At this point, it is apparent why the Exp mapping is not injective. Since the Log mapping seeks the smallest-norm solution, we take $\|v\|_2 = \arccos \langle y, x \rangle$. Finally, solving for the direction of $v$,

$$\mathrm{Log}_x(y) = v = \frac{\|v\|_2}{\sin \|v\|_2}(y - \cos(\|v\|_2)x) = \frac{\arccos \langle y, x \rangle}{\sqrt{1 - \langle y, x \rangle^2}}(y - \langle y, x \rangle x). \tag{31}$$

## B.2 Determinant of Jacobian of exponential map

**Proposition 3.** *Let $x \in \mathbb{S}^n$ and $v \in \mathcal{T}_x\mathbb{S}^n$. Then,*

$$\left| \frac{\partial \mathrm{Exp}_x(v)}{\partial v} \right| = \left( \frac{\sin \|v\|_2}{\|v\|_2} \right)^{n-1}$$

*Proof.* We adapt the proof of Nagano et al. (2019) for determinant of the exponential map on a hyperbolic space. We start by noting that $\frac{\partial \mathrm{Exp}_x(v)}{\partial u}$ is a function, which acts from $\mathcal{T}_x\mathbb{S}^n$ to $\mathcal{T}_{\mathrm{Exp}_x(v)}\mathbb{S}^n$. We are free to select an orthonormal basis in which we compute the Jacobian $\left| \frac{\partial \mathrm{Exp}_x(v)}{\partial u} \right|$. Hence, we choose a basis by including $\overline{v} = \frac{v}{\|v\|}$, and any other set of orthogonal unit vectors in $\mathcal{T}_x\mathbb{S}^n$: $\{\overline{v}, v'_1, \ldots, v'_{n-1}\}$. Next, we compute a set of directional derivatives $w.r.t.$ to each of the basis vectors.

$$\begin{aligned}
\mathrm{d}\,\mathrm{Exp}_x(\overline{v}) &= \frac{\partial}{\partial \epsilon}\bigg|_{\epsilon=0} \mathrm{Exp}_x(v + \epsilon\overline{v}) \\
&= \frac{\partial}{\partial \epsilon}\bigg|_{\epsilon=0} \left[ \cos(\|v\|_2 + \epsilon)x + \sin(\|v\|_2 + \epsilon)\frac{v + \epsilon\overline{v}}{\|v + \epsilon\overline{v}\|} \right] \\
&= -\sin(\|v\|_2)x + \cos(\|v\|_2)\overline{y}.
\end{aligned}$$

Here we used $\frac{\partial}{\partial \epsilon}\frac{v+\epsilon\overline{v}}{\|v+\epsilon\overline{v}\|} = 0$ since the norm of the vector does not change $w.r.t.$ $\epsilon$

The norm of this directional derivative $w.r.t.$ $\overline{v}$ is equal to 1:

$$\|-\sin(\|v\|_2)x + \cos(\|v\|_2)\overline{v}\|_2 = \sqrt{\sin(\|v\|_2)^2 + \cos(\|v\|_2)^2} = 1$$

Next, we calculate the directional derivatives $w.r.t.$ the other basis vectors $v'_1, \ldots, v'_{n-1}$:

$$\begin{aligned}
\mathrm{d}\,\mathrm{Exp}_x(v'_k) &= \frac{\partial}{\partial \epsilon}\bigg|_{\epsilon=0} \mathrm{Exp}_x(v + \epsilon v'_k) \\
&= \frac{\partial}{\partial \epsilon}\bigg|_{\epsilon=0} \left[ \cos(\|v\|_2)x + \sin(\|v\|_2)\frac{v + \epsilon v'_k}{\|v\|_2} \right] \\
&= \frac{\sin \|v\|_2}{\|v\|_2}v'_k.
\end{aligned}$$

The norm of the directional derivative $w.r.t.$ $v'_k$ is thus $\left\|\frac{\sin\|v\|_2}{\|v\|_2}v'_k\right\|_2 = \frac{\sin\|v\|_2}{\|v\|_2}$.

Finally, the determinant is the product of norms of directional derivatives since the chosen basis is orthonormal:

$$\det\left( \frac{\partial \mathrm{Exp}_x(v)}{\partial v} \right) = \left( \frac{\sin \|v\|_2}{\|v\|_2} \right)^{n-1}$$

$\square$

Note that $n$ here is the dimension of the sphere and not the dimension of the ambient space.

## B.3 Wasserstein-2 distances and interpolation

Mallasto & Feragen (2018) provides an analytical expression for a pullback Wasserstein-2 squared distance between two wrapped normal distributions defined over the tangent bundle, which we can directly adapt to calculate pullback Wasserstein-2 squared distance between two $\mathcal{WN}_\beta$ distributions $p_1 = \mathcal{WN}_\beta(\mu_1, \Sigma_1)$ and $p_2 = \mathcal{WN}_\beta(\mu_2, \Sigma_2)$ transported to a shared tangent plane at some point $\mu$:

$$\mathcal{W}_2^2(p_1, p_2) = d_\mathcal{M}^2(\mu_1, \mu_2) + \mathcal{W}_2^2(PT_{\mu_1 \to \mu}\mathcal{N}_\beta(0, \Sigma_1), PT_{\mu_2 \to \mu}\mathcal{N}_\beta(0, \Sigma_2)) \tag{32}$$

For pole parametrization, when tangent $\beta$-Gaussians are already defined over a common tangent space the expression simplifies to

$$\mathcal{W}_2^2(p_1, p_2) = d_\mathcal{M}^2(\mu_1, \mu_2) + \frac{R^2}{N + \frac{2(2-\beta)}{1-\beta}}\mathfrak{B}^2(\tilde{\Sigma}_1, \tilde{\Sigma}_2), \tag{33}$$

where $\mathfrak{B}^2(A, B) \coloneqq \mathrm{Tr}\left(A + B - 2\left(A^{1/2}BA^{1/2}\right)^{1/2}\right)$ (Martins et al., 2022; Gelbrich, 1990).

In Figure 8, we plot the pullback Wasserstein-2 interpolation between two wrapped $\beta$-Gaussian distributions.

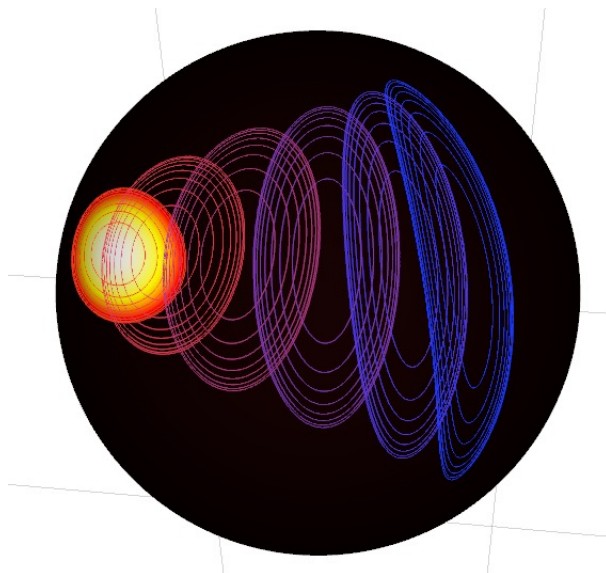

**Figure 8:** Pullback Wasserstein-2 interpolation between two wrapped $\beta$-Gaussians over the $\mathbb{S}^2$.

## B.4 Fitting data with $L^\times$ loss

In Figure 9, we provide additional visualization for tangent and wrapped losses for higher dimensionality ($\mathbb{S}^{29}$), where the influence of wrapping is higher.

Additionally, we experiment with fitting a dataset of points on $\mathbb{S}^2$ using the Fench-Young losses. To fit the wrapped $\beta$-Gaussian to the dataset $X$ we use the tangent $\ell^\times$ loss from Eq. (11), and compare it to $L^\times$ loss with Jacobian correction (Eq. (15)). For the simple experiment, we create a dataset of size 10000 by sampling from a true wrapped $\beta$-Gaussian (initialized randomly to have non-isotropic covariance). We initialize the wrapped $\beta$-Gaussian with a random location and scale parameters and optimize the location parameter with Riemannian Adam (Becigneul & Ganea, 2019), with the scale parameterized as in Eq. (16), with learning rate 0.01 (other hyperparameters are default) for 5000 iterations.

**Results** In Figure 10, we show that by optimizing the Fenchel-Young loss, we can successfully recover the true location and covariance parameters. Both tangent loss and wrapped loss recover true parameters, while the tangent loss converges faster in our experiments. In this experiment, the accuracy of finding $\mu$,

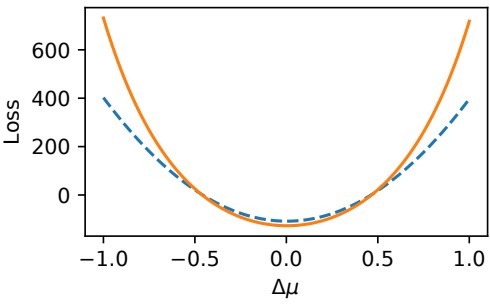 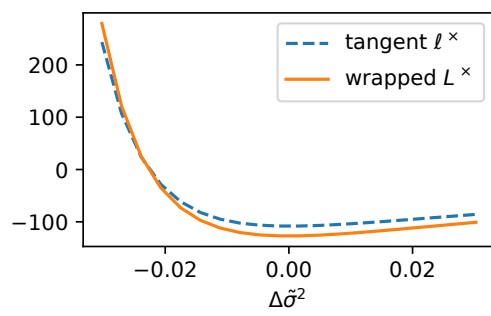

**(a)** Loss *w.r.t.* shifting the location $\mu$ from the true location: $\hat{\mu} = \text{Log}_\mu(PT_{p \to \mu}(\Delta\mu, 0))$.

**(b)** Loss *w.r.t.* shifting the $\tilde{\sigma}^2$ of the $\mathcal{WN}_\beta(\mu, \tilde{\sigma}^2 I)$.

**Figure 9:** Visualization of wrapped and tangent Fenchel-Young losses *w.r.t.* to (a) location, and (b) scale shift from the true parameters of $\mathcal{WN}_\beta(\mu, \tilde{\sigma}^2 I)$, where $\mu$ is random on the $\mathbb{S}^{29}$, and the trace of $\text{Cov}_\mu = 0.3$. Losses are averaged over 10000 samples from the true $\mathcal{WN}_\beta$ distribution ($\beta = 0.9$).

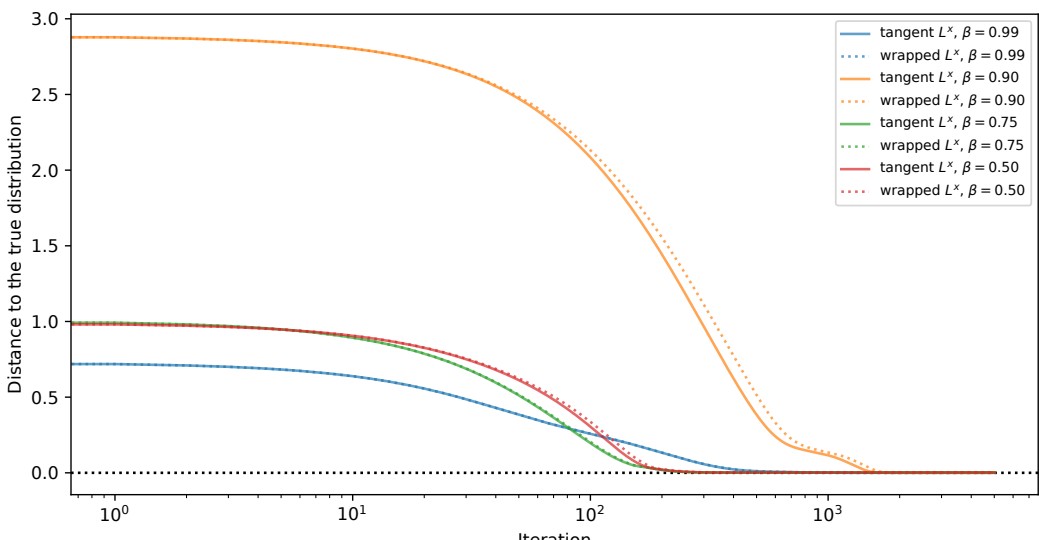

**Figure 10:** Fitting data on $\mathbb{S}^2$. For different $\beta$ parameters, we fit the wrapped $\beta$-Gaussian on the samples from the true wrapped $\beta$-Gaussian with the same $\beta$. For each run, we plot pullback $\mathcal{W}_2^2$ distance (Appendix B.3) between true wrapped $\beta$-Gaussian and the fitted one (with the same $\beta$). Solid colored lines represent fitting with tangent $\ell^\times$, and dashed colored lines represent fitting with wrapped $L^\times$ (with Jacobian correction). Different colors represent different $\beta$. On the $x$ axis is the number of optimization steps.

$\tilde{\Sigma}$ is comparable for moment matching and FY loss optimization ($< 0.02$ pullback $\mathcal{W}_2^2$ error, as defined in Appendix B.3). When available, the iterative process based on Karcher mean is empirically preferrable.

## C Extrinsic parametrization of covariances on the sphere: Proof of Proposition 2

For full-rank covariance $\tilde{\Sigma}$ defined in the ambient space, the following result allows us to calculate pseudo-determinants and pseudoinverses efficiently. For narrative purposes, we split the proposition into separate results for the full-rank and rank-deficient cases.

**Lemma 2.** *Let $S$ be a $n$-by-$n$ positive definite scale parameter and $P = (I - xx^\top)$ be the projection operator onto the hyperplane orthogonal to the unit vector $x$. We have:*

$$(i) \qquad |PSP|_+ = |S| \cdot x^\top S^{-1} x,$$

$$(ii) \qquad (PSP)^+ = S^{-1} - \frac{S^{-1}xx^\top S^{-1}}{x^\top S^{-1} x},$$

$$(iii) \qquad v^\top (PSP)^+ v = v^\top S^{-1} v \frac{(x^\top S^{-1} v)^2}{x^\top S^{-1} x}.$$

*Proof.* (i) For any two matrices $A, B$, $AB$ and $BA$ have the same nonzero eigenvalues. Therefore, $|PSP|_+ = |PPS|_+ = |PS|_+$, since projection operators are idempotent. Following Grossmann (2022), since $PS$ has rank $n - 1$, its adjugate $\mathrm{adj}(PS)$ has a single nonzero eigenvalue equal to $|PS|_+$ and therefore:

$$|PS|_+ = \mathrm{tr}(\mathrm{adj}(PS)) = \mathrm{tr}(\mathrm{adj}(S)\,\mathrm{adj}(P)) = |S|\,\mathrm{tr}(S^{-1}\,\mathrm{adj}(P)).$$

Since $P$ has $n - 1$ non-null eigenvalues equal to 1, $\mathrm{adj}(P) = xx^\top$ and so $|PSP|_+ = |S|x^\top S^{-1} x$.

(ii) Let $M = S^{1/2}P$ be a square root of $PSP$, so $M^\top M = PSP$. For any matrix, $(M^\top M)^+ = M^+(M^+)^\top$, so we just need to find $M^+$. Remark that

$$M = S^{1/2}P = S^{1/2} - S^{1/2}xx^\top = S^{1/2} + cd^\top,$$

with $c = -S^{1/2}x$ and $d = x$. $M$ is thus a rank-one update to $S^{1/2}$. Meyer (1973) categorizes such updates depending on whether $c$ and $d$ are in row (column) space of $S^{1/2}$ and on the value $\beta = 1 + c^\top S^{-1/2}d = 1 - x^\top S^{1/2}S^{-1/2}x = 1 - x^\top x = 0$. Since $S$ is full-rank, its row and column spaces contain all non-null vectors, leaving us under the scope of Meyer (1973, Theorem 6) which reads:

$$\begin{aligned}
M^+ = \ &S^{-1/2} \\
&- xx^\top S^{-1/2} \\
&- \frac{1}{x^\top S^{-1} x}S^{-1}xx^\top S^{-1/2} \\
&+ \frac{-x^\top S^{-1} x}{x^\top S^{-1} x}(-xx^\top S^{-1/2})
\end{aligned}$$

The second and fourth term cancel out, leaving

$$M^+ = S^{-1/2} - \frac{1}{x^\top S^{-1} x}S^{-1}xx^\top S^{-1/2} = S^{-1/2}(I - zz^\top),$$

where we define $z = \frac{S^{-1/2}x}{\|S^{-1/2}x\|}$ such that $(I - zz^\top)$ is an orthogonal projection, and thus idempotent. Finally,

$$\begin{aligned}
(PSP)^+ = M^+(M^+)^\top &= S^{-1/2}(I - zz^\top)S^{-1/2} \\
&= S^{-1} - \frac{S^{-1}xx^\top S^{-1}}{x^\top S^{-1} x}.
\end{aligned}$$

Pre- and post-multiplying by any vector $v$ gives the relationship in (iii). $\qquad\square$

In practice, covariances are usually enforced to be full rank in parametrization. Nevertheless, for completeness we derive in the next proposition the corresponding expressions for low-rank $\tilde{\Sigma}$.

**Lemma 3.** *Let $S$ be an $n$-by-$n$ low-rank matrix and $R = I - S^+(S^+)^\top$ be the orthogonal projection onto the kernel of $S$. Let $x$ be a unit vector and $P = I - xx^\top$ be its tangent projection, and let $\beta = x^\top R x$.*

*If $\beta = 0$ then Lemma 2 applies, replacing determinants/inverses with pseudo versions.*

If $\beta > 0$, we have

$$(i) \qquad |PSP|_+ = |S|_+ \cdot \beta,$$

$$(ii) \qquad (PSP)^+ = \left(I - \frac{1}{\beta}Rxx^\top\right)S^+\left(I - \frac{1}{\beta}Rxx^\top\right)^\top.$$

*Proof.* For the $\beta = 0$ case, the condition implies $x \in \text{Span}\,S$. As both $P$ and $S$ have no action on the kernel of $S$, we can apply a rotation that moves those dimensions to the end, and perform all calculations on the full rank top-left block.

When $\beta \neq 0$, things dramatically change.

(i) We make use of the limit definition of pseudodeterminants:

$$|A|_+ = \lim_{\alpha \to 0} |A + \alpha I|/\alpha^{n - \text{rank}\,A}. \tag{34}$$

In the full-rank case, $\text{rank}\,PSP = n - 1$. In general $\text{rank}\,PSP = \text{rank}\,S - 1$ only if $x \in \text{Span}\,S$, which we have ruled out, otherwise $\text{rank}\,PSP = \text{rank}\,S$. Since $P$ is a projection onto a subspace of dimension $n - 1$, we have $P = UU^\top$ where $U \in \mathbb{R}^{n,n-1}$ and $U^\top U = I_{n-1}$. $U$ has orthogonal columns, so the spectrum of $PSP$ is equal to the one of $U^\top PSPU = U^\top SU$. We therefore have

$$
\begin{aligned}
|PSP|_+ &= |U^\top PSPU|_+ \\
&= \lim_{\alpha \to 0} |U^\top PSPU + \alpha I|/\alpha^{(n-1)-\text{rank}\,S} \\
&= \lim_{\alpha \to 0} |PSP + \alpha P|_+/\alpha^{(n-1)-\text{rank}\,S} \qquad \text{(left and right multiply by } U \text{ and } U^\top) \\
&= \lim_{\alpha \to 0} |P(S + \alpha I)P^\top|_+/\alpha^{(n-1)-\text{rank}\,S} \\
&= \lim_{\alpha \to 0} |S + \alpha I| \cdot x^\top(S + \alpha I)^{-1}x/\alpha^{(n-1)-\text{rank}\,S} \\
&= |S|_+ \lim_{\alpha \to 0} \alpha x^\top(S + \alpha I)^{-1}x \\
&= |S|_+ x^\top Rx,
\end{aligned}
\tag{35}
$$

where the final step comes from noticing that the eigenvalues of $\alpha(S + \alpha I)^{-1}$ are of the form $\hat{\lambda}_j = \frac{\alpha}{\alpha + \lambda_j}$, where $\lambda_j$ are the eigenvalues of $S$. In the limit, $\hat{\lambda}_j = 1$ if $\lambda_j = 0$ and 0 otherwise, therefore $\lim_{\alpha \to 0} \alpha(S + \alpha I)^{-1} = R$, finishing this part of the proof.

(ii) We once again turn to Meyer (1973), finding ourselves under the auspices of their Theorem 3, since $\beta \neq 0$. We define the same symbols as in the previous proof, except $S^{-1/2}$ is replaced by $(S^{1/2})^+$, and therefore the terms that become an identity matrix in the full-rank case now become $(S^{1/2})^+(S^{1/2}) = I - R$. The $q_1$ term in Meyer's theorem is null. Carrying out the calculation leads to

$$M^+ = \left(I - \frac{1}{\beta}Rxx^\top\right)(S^{1/2})^+.$$

Using again $(PSP)^+ = M^+(M^+)^\top$ yields the desired result. Efficient Mahalanobis distances by calculating $vM^+$ without materializing $M^+$ are left as an exercise. $\qquad\square$

Combining the two lemmas yields a proof of the proposition.

## D  The wrapped $\beta$-Gaussian over Rotation Matrices

Our construction is not limited to spheres but is applicable and useful in other manifolds, as long as we know the injectivity radius, the Log and Exp mappings, and parallel transport.

In this section, we instantiate wrapped $\beta$-Gaussians over the manifold of rotation matrices, also known as the special orthogonal group $SO(n)$.

### D.1 Preliminary: properties of skew-symmetric matrices

The set of real skew-symmetric matrices is defined as

$$\text{Skew}(n) := \{A \in \mathbb{R}^{n \times n} : A^\top = -A\}.$$

The next result recaps some classical results about skew-symmetric matrices.

**Lemma 4** (Properties of skew-symmetric matrices)**.** *Let $A \in \text{Skew}(n)$ with $m = \lfloor n/2 \rfloor$.*

*(i) The nonzero eigenvalues of $A$ are purely imaginary and come in pairs $\lambda_{+j} = i\theta_j, \lambda_{-j} = -i\theta_j$.*

*(ii) The eigenvectors of $A$ have the form $p_{\pm j} = \frac{1}{\sqrt{2}}q_{+j} \pm \frac{1}{\sqrt{2}}iq_{-j}$, with $\|q_{\pm j}\| = 1$ for all $j$, and $\langle q_s, q_t \rangle = 0$ for any $s \neq t$.*

*(iii) $A$ admits the canonical decomposition*

$$A = \sum_{j=1}^{m} \theta_j (q_{+j}q_{-j}^\top - q_{-j}q_{+j}^\top). \tag{36}$$

*(iv) For even $n$, forming the matrix $Q$ with columns $(q_{+1}, q_{-1}, \ldots, q_{+m}, q_{-m})$, and the block-diagonal matrix*

$$B = \begin{pmatrix} 0 & \theta_1 & & & & & \\ -\theta_1 & 0 & & & & & \\ & & 0 & \theta_2 & & & \\ & & -\theta_2 & 0 & & & \\ & & & & \ddots & & \\ & & & & & 0 & \theta_m \\ & & & & & -\theta_m & 0 \end{pmatrix}, \tag{37}$$

*$Q$ is orthogonal and we have $A = QBQ^\top$. For odd $n$, extend $B$ with an additional zero row and column, and $Q$ with a unit vector from the null space of $A$.*

*(v) $A$ admits a (potentially truncated) singular value decomposition with singular triplets $\{(\theta_j, q_{+j}, q_{-j})\}_{j=1,\ldots,m} \cup \{(\theta_j, q_{-j}, -q_{+j})\}_{j=1,\ldots,m}$*

*(vi) $\exp(A)$ is a special orthogonal matrix and is equal to $Q\exp(B)Q^\top$.*

*Proof.* A proof of properties (i)–(iv) can be found in Gantmacher (1959, §IX.12). For completeness, since the proofs are compact, we re-derive them here.

(i) The complex eigenvalues and eigenvectors of any real matrix come in conjugate pairs: if $Ap = \lambda p$, taking conjugates on both sides gives $\bar{\lambda}\bar{p} = \bar{A}\bar{p} = A\bar{p}$. For skew-symmetric A, we have on one hand

$$\langle Ap, p \rangle = \langle \lambda p, p \rangle = \bar{\lambda}\langle p, p \rangle$$

and on the other hand

$$\langle Ap, p \rangle = p^H A^H p = p^H A^\top p = p^H(-Ap) = p^H(-\lambda p) = -\lambda \langle p, p \rangle.$$

Therefore $\lambda$ is purely imaginary, and for convenience we index the pairs as $\lambda_{\pm j} = \pm i\theta_j$ for $\theta_j \in \mathbb{R}$.

(ii) If $p$ is an eigenvector of $A$ associated with a nonzero eigenvalue, then so is $\bar{p}$ and since their eigenvalues are distinct they must be orthogonal. Writing $p = b + id$,

$$\langle \bar{p}, p \rangle = p^\top p = (b + id)^\top(b + id) = (b^\top b - d^\top d) + i(b^\top d + d^\top b). \tag{38}$$

It follows that $b^\top b = d^\top d$. But since eigenvectors have length 1, we have

$$1 = \|p\|^2 = b^\top b + d^\top d.$$

Combining this with the fact that the real part of Eq. (38) is zero, we get $\|b\| = \|d\| = \frac{1}{\sqrt{2}}$, so we set $b = \frac{1}{\sqrt{2}}q_+$ and $d = \frac{1}{\sqrt{2}}q_-$ where $\|q_\pm\| = 1$. From setting the imaginary part of Eq. (38) to zero we get $b^\top d = 0$ so $q_+^\top q_- = 0$. For two distinct $j, k$ we use orthogonality to obtain:

$$0 = 2\langle p_{+j}, p_{+k}\rangle = q_{+j}^\top q_{+k} - q_{-j}^\top q_{-k} + i q_{+j}^\top q_{-k} - i q_{-j}^\top q_{+k},$$
$$0 = 2\langle p_{-j}, p_{+k}\rangle = q_{+j}^\top q_{+k} + q_{-j}^\top q_{-k} + i q_{+j}^\top q_{-k} + i q_{-j}^\top q_{+k}.$$

Adding and subtracting the two relationships reveals the orthogonality of $q_{\pm j}$, $q_\pm k$, and $q_\mp k$.

(iii) First, note that, if $p = b + id$,

$$pp^H - \bar{p}\bar{p}^H = (b + id)(b - id)^\top - (b - id)(b + id)^\top = 2i(db^\top - bd^\top). \tag{39}$$

We rearrange the eigendecomposition:

$$
\begin{aligned}
A &= \sum_j \left[ i\theta_j p_{+j} p_{+j}^H - i\theta_j p_{-j} p_{-j}^H \right] \\
&= \sum_j \left[ i\theta_j (p_{+j} p_{+j}^H - p_{-j} p_{-j}^H) \right] \\
&= \sum_j \left[ \frac{i\theta_j}{2} (2i(q_{-j} q_{+j}^\top - q_{+i} q_{-j}^\top)) \right] \\
&= \sum_j \left[ \theta_j (q_{+j} q_{-j}^\top - q_{-i} q_{+j}^\top) \right].
\end{aligned}
$$

(iv) and (iv). Both follow from (iii) and the fact that all $q_{\pm j}$ are unit length and all distinct-index pairs are orthogonal.

(vi) We may start from the decomposition $A = QBQ^\top$. Note that $A^k = QB^kQ^\top$ for natural powers, since $Q^\top Q = I$. Since the matrix exponential is defined through a power series, we have $\exp(A) = Q\exp(B)Q^\top$. $\exp(B)$ is a block-diagonal matrix with each block corresponding to a rotation by $\theta_j$: $\begin{pmatrix} \cos\theta_j & \sin\theta_j \\ -\sin\theta_j & \cos\theta_j \end{pmatrix}$, and therefore is orthogonal with determinant 1. Therefore so is the product of $QBQ^\top$. $\qquad\square$

## D.2 Standard toolbox for $SO(n)$

$SO(n)$ can be seen as an embedded manifold of $\mathbb{R}^{n \times n}$ defined as follows:

$$SO(n) := \{X \in \mathbb{R}^{n \times n} : X^\top X = I_n, \det(X) = 1\}.$$

The tangent space is characterized as

$$\mathcal{T}_X SO(n) = \{V \in \mathbb{R}^{n \times n} : V^\top X + X^\top V = 0\},$$

which can be seen by differentiating the orthogonality constraint $X^\top X = I_n$ in the ambient space.

The identity matrix $I_n$ is a distinguished element of $SO(n)$. Its tangent space $\mathcal{T}_I SO(n)$ is the space of skew-symmetric matrices $\text{Skew}(n)$, which can be seen to be a vector space of dimension $n(n-1)/2$, and in this context it is often called the Lie algebra $\mathfrak{so}(n)$. At the identity matrix, the Exp and Log mappings between $SO(n)$ and $\mathcal{T}_I SO(n) = \mathfrak{so}(n)$ are exactly the matrix exponential and matrix logarithm:

$$\text{Exp}_I(A) = \exp(A), \qquad \text{Log}_I(Y) = \log(Y).$$

Now remark that any rotation $X \in SO(n)$ is also an invertible linear operator that maps $I_n$ to $X$. Moreover, if $A \in \mathcal{T}_I SO(n)$ then $X^\top X A + A^\top X^\top X = A + A^\top = 0$, therefore $XA \in \mathcal{T}_X SO(n)$. Since orthogonal matrices

are isometries, we can rotate the ambient space by multiplying by $X^{-1}$ in order to identify $\mathcal{T}_X SO(n)$ with $\mathfrak{so}(n)$ (since the rotation takes $X$ to $I_n$) and thus we have

$$\mathcal{T}_X SO(n) = \{XA : A \in \text{Skew}(n)\},$$

and we can now compute:

$$\text{Exp}_X(A) = X \exp(X^\top A), \qquad \text{Log}_X(Y) = X \log(X^\top Y).$$

From the same observation it also follows that $PT_{X \to Y} = YX^{-1}$. *Remark.* Since all orthogonal matrices are invertible, all tangent spaces are isomorphic to $\mathfrak{so}(n)$. It is common in practice to identify them and avoid some of the redundant parallel transports. With this representation we would write $\widetilde{\text{Exp}}_X(A) = X \exp(A)$ and $\widetilde{\text{Log}}_X(Y) = \log(X^\top Y)$.

The matrix exponential and logarithm are not univerally inverses of each other; the following result gives the injectivity radius.

**Proposition 4.** *The injectivity radius of $SO(n)$ is $\pi\sqrt{2}$.*

*Proof.* We recap the proof given by Axen (2023), with a few more details. By symmetry we need only focus on the injectivity of $\text{Exp}_I = \exp$. Let $A = P\Lambda P^H$ be the spectral decomposition $A \in \text{Skew}(n)$, with eigenvalues of the form $\lambda_{+j} = \pm i\theta_j$. $A$ is in the injectivity domain iff $A = \log(\exp(A))$, or equivalently $\lambda = \log(\exp(\lambda))$ for all eigenvalues of $A$. The injectivity domain therefore constrains $\theta_j \in (-\pi, \pi)$. Let $r$ denote the radius of the largest ball contained in the injectivity domain, and notice that $\|A\|_F^2 = \text{tr}(A^\top A) = 2\sum_j \theta_j^2$. Consider the matrix $A_1$ defined by $\theta_1 = \pi$ and all other $\theta_j = 0$: this matrix is just on the boundary of the injectivity domain and $\|A_1\|_F = \pi\sqrt{2}$, so $r \leq \pi\sqrt{2}$. On the other hand, for an arbitrary $A$ we have $\|A\|_F^2 \geq 2\theta_j$ for any $j$, and so $\|A\|_F < \pi\sqrt{2}$ implies $|\theta_j| < \pi$ for any $j$, so $r \geq \pi\sqrt{2}$. $\qquad\square$

Finally, for completeness, we give an explicit orthonormal basis of $\text{Skew}(n)$, allowing us to parametrize wrapped distributions in this space.

**Proposition 5.** *Let $e_{jk} \in \mathbb{R}^{n \times n}$ be an indicator matrix with all elements set to zero except the element at index $j, k$, which is set to one. Let $D$ denote the linear operator $D : \mathbb{R}^{n(n-1)/2} \to \text{Skew}(n) \subset \mathbb{R}^{n \times n}$ with columns[a]*

$$D_{jk} = \frac{1}{\sqrt{2}} \left( e_{jk} - e_{jk}^\top \right).$$

*Then, $D$ is an orthogonal matrix.*

---
[a]Given a vector $v \in \mathbb{R}^{n(n-1)/2}$, $Dv = (V - V^\top)/\sqrt{2}$ where $V$ is a lower triangular matrix with the lower triangle given by the elements of $v$.

*Proof.* Verify that the columns of $D$ are by construction skew-symmetric, pairwise orthogonal, and have unit norm, as: $\|D_{ij}\|_F^2 = 2(\frac{1}{\sqrt{2}})^2 = 1$. They therefore form an orthonormal basis of $\text{Skew}(n)$ and we have $D^\top D = I_{n(n-1)/2}$. $\qquad\square$

We can therefore construct a wrapped $\beta$-Gaussian on $SO(n)$ as:

$$X \sim W\mathcal{N}_\beta(M, \Sigma) \iff X = \widetilde{\text{Exp}}_M(V) = M\exp(V), V = Dv, v \sim \mathcal{N}_\beta(0, \Sigma).$$

### D.3 Jacobian Log-Determinant

To compute the (log)-density of $W\mathcal{N}_\beta$ random variables, we apply the change of density formula. Multiplication by an orthogonal matrix is an isometry, and we have shown that so is the embedding $D$. It remains to calculate the change of volume induced by the matrix exponential.

**Proposition 6.** *Let $A \in \text{Skew}(n)$, $m = \lfloor n/2 \rfloor$, and denote the eigenvalues of $A$ as $\lambda_{\pm j} = \pm i\theta_j$, for $j = 1, \ldots, m$, and for odd $n$ an additional $\lambda_0 = \theta_0 = 0$. Let $J$ denote the Jacobian of the matrix exponential at $A$, i.e., $J_{ij,kl} = \frac{\partial(\exp A)_{ij}}{\partial A_{kl}}$. Then,*

$$\det J = \prod_{j=\gamma}^{m} \prod_{k=1}^{m} \text{sinc}^2\left(\frac{\theta_j - \theta_k}{2}\right) \text{sinc}^2\left(\frac{\theta_j + \theta_k}{2}\right). \tag{40}$$

*where the range of $j$ (but not of $k$) starts at $\gamma = 0$ if $n$ is odd, and at $\gamma = 1$ otherwise,*

*Proof.* First, we note the result is similar to the case of Grassmanians given by Chevallier et al. (2022, eq. 4.4). We proceed to give a complete proof.

Per Higham (2008, Theorem 3.9), as exp is analytic, the eigenvalues of $J$ are

$$\nu_{st} = \begin{cases} \frac{\exp \lambda_s - \exp \lambda_t}{\lambda_s - \lambda_t}, & s \neq t, \\ \exp \lambda_s, & s = t. \end{cases}$$

For a general characterization of the spectral decomposition of $J$ see Magnus et al. (2021).

The eigenvalues of $J$ are complex, but we next show that for skew-symmetric $A$ the determinant is always real. Let us assume for now that $n$ is even. The eigenvalues of a skew-symmetric matrix are purely imaginary and come in conjugate pairs, so we may use the indexing $\lambda_{+j} = i\theta_j$ and $\lambda_{-j} = -i\theta_j$ for $j = 1, \ldots, n/2$. In this notation, the indices $s$ and $t$ take values in $\{\pm 1, \ldots, \pm m\}$.

We notice that the eigenvalues of the Jacobian can be grouped two by two:

$$\begin{aligned}
\nu_{+j+k}\nu_{-j-k} &= \frac{\exp i\theta_j - \exp i\theta_k}{i\theta_j - i\theta_k} \cdot \frac{\exp -i\theta_j - \exp -i\theta_k}{-i\theta_j + i\theta_k} \\
&= \frac{2 - \exp i(\theta_j - \theta_k) - \exp -i(\theta_j - \theta_k)}{i(\theta_j - \theta_k) \cdot (-i)(\theta_j - \theta_k)} \\
&= \frac{2 - 2\cos(\theta_j - \theta_k)}{(\theta_j - \theta_k)^2}
\end{aligned}$$

$$\begin{aligned}
\nu_{+j-k}\nu_{-j+k} &= \frac{\exp i\theta_j - \exp -i\theta_k}{i\theta_j + i\theta_k} \cdot \frac{\exp -i\theta_j - \exp i\theta_k}{-i\theta_j - i\theta_k} \\
&= \frac{2 - \exp i(\theta_j + \theta_k) - \exp -i(\theta_j + \theta_k)}{i(\theta_j + \theta_k) \cdot (-i)(\theta_j + \theta_k)} \\
&= \frac{2 - 2\cos(\theta_j + \theta_k)}{(\theta_j + \theta_k)^2}
\end{aligned}$$

Using the fact that $2 - 2\cos\phi = 4\sin^2\frac{\phi}{2}$ we rewrite both terms above in the form $\text{sinc}^2\left(\frac{\theta_i \pm \theta_j}{2}\right)$, yielding the desired result. We point out that further simplification can be obtained by recognizing symmetry ($\nu_{st} = \nu_{ts}$) and noticing that the diagonal terms cancel out because $\nu_{+i+i}\nu_{-i-i} = 1$. Calculating the (log) determinant with a minimum number of operations can be achieved by considering only the upper triangle.

In the case of odd $n$, there is an additional unpaired zero eigenvalue which we index with $s = 0$. Then, we have to consider the additional entries:

$$\nu_{00} = \exp 0 = 1$$

$$\nu_{0t} = \nu_{t0} = \frac{\exp \lambda_t - 1}{\lambda_t}.$$

Using a similar calculation to above we get that $\nu_{0+i}\nu_{0-i} = \nu_{+i0}\nu_{-i0} = \text{sinc}^2\left(\frac{\theta_i}{2}\right)$, which for brevity we can absorb by extending the range of $j$ in the product. $\qquad\square$

**Numerical details.** By continuity, $\text{sinc}(0) = 1$. For stable computation directly in log-domain, use

$$\log \text{sinc}(\pi x) = -\log \Gamma(1-x) - \log \Gamma(1+x). \tag{41}$$

In general, just like in the spherical case, the log-determinant can go to $-\infty$. However, Proposition 4 implies that within the injectivity domain of $\mathcal{T}_I SO(n)$, we have $|\theta| < \pi$, and so all the sinc terms in Eq. (40) take arguments in $(-\pi, \pi)$ and are therefore finite.

**Computing the rotation angles.** Learning with a wrapped distribution over $SO(n)$ requires evaluating the determinant of the Jacobian (which requires the eigenvalues of $A$) as well as calculating the matrix exponential itself. This is in contrast to applications where just the matrix exponential is needed, where truncated series approximations are widely applicable. Obtaining an eigendecomposition of $A$ would allow us to compute both quantities needed; however, complex decompositions are more expensive. The following propositions shows how to efficiently use a SVD of $A$ to obtain both the rotation angles $\theta$ and the value of the matrix exponential.

**Proposition 7** (Computation). *Let $A \in \text{Skew}(n)$ and $A = USV^\top$ be a singular value decomposition. Then,*

$$\exp(A) = U \cos(S) U^\top + U \sin(S) V^\top.$$

*Proof.* Assume for now that $n$ is even. From Lemma 4, item (vi), we have

$$\exp(A) = \sum_j Q_j \exp(B_j) Q_j^\top, \text{ where } Q_j = \begin{pmatrix} q_{+j} \\ q_{-j} \end{pmatrix}, \text{ and } \exp(B_j) = \begin{pmatrix} \cos\theta_j & \sin\theta_j \\ -\sin\theta_j & \cos\theta_j \end{pmatrix}. \tag{42}$$

Explicitly calculating one such rank-2 term gives:

$$\begin{aligned} Q_j \exp(B_j) Q_j^\top = {}& \cos\theta_j (q_{+j} q_{+j}^\top + q_{-j} q_{-j}^\top) \\ & + \sin\theta_j (q_{+j} q_{-j}^\top - q_{-j} q_{+j}^\top). \end{aligned}$$

From Lemma 4 item (v), one possible SVD is given by the triplets $(\theta_j, q_{+j}, q_{-j}), (\theta_j, q_{-j}, -q_{+j})$. A straightforward calculation shows that the part of $U \cos(S) U^\top + U \sin(S) V^\top$ corresponding to the two copies of $\theta_j$ is exactly $Q_j \exp(B_j) Q_j^\top$. While SVD is only unique up to a simultaneous sign flip of a singular vector pair $(y, v)$, the rank-one terms are invariant, since $(-u)(-u)^\top = uu^\top$ and $(-u)(-v)^\top = uv^\top$. Therefore, the desired expression holds for any SVD. If $n$ is odd, there is an additional row $q_0$ in the canonical decomposition, with a corresponding 1-d block $B_0 = (0)$ so $\exp(B_0) = (1)$ and $\exp(A)$ has an additional rank-1 term $q_0 q_0^\top$. In this case, a complete SVD of $A$ has an unpaired zero singular triplet $(\theta_0 = 0, \pm q_0, \pm q_0)$. Regardless of the sign choice, since $\sin(0) = 0$ and $\cos(0) = 1$, we get an additional term $q_0 q_0^\top$ as expected. Other than the sign ambiguity, there is no ambiguity about the direction of the singular vector corresponding to the unpaired singular value, since it must be orthogonal to the $2m$ distinct $q_{\pm j}$ vectors, even if some of them may correspond to other zero singular values. $\qquad \square$

# E Algorithms for $\mathbb{S}^n$

Here we provide the algorithm for the parallel transport for the hypersphere.

---
**Algorithm 1:** Parallel transport for $\mathbb{S}^{n-1}$ embedded in $\mathbb{R}^n$.

---
**Input:** $p, x \in \mathbb{R}^n$, $\|p\|_2 = 1 \|x\|_2 = 1$; $x \neq -p$; $u$: the point on the $\mathcal{T}_p \mathbb{S}^n$

$z' = (p + x)$;

$z = \frac{z'}{\|z'\|}$ ;                                    // $z$ defines reflection plane $R_{\frac{\mu+p}{\|\mu+p\|}}$

**return** $u - 2\langle u, z\rangle z$ *(rotated vector)*

---

