# OpenReview forum: "Wrapped $\beta$-Gaussians with compact support for exact probabilistic modeling on manifolds"
_TMLR — Accepted by TMLR_

### Review · Reviewer_LAzL · 2023-07-19

**Summary Of Contributions:**

In this paper, the authors propose a novel family of distributions defined on Riemannian manifolds named wrapped $\beta$-Gaussians, which can be seen as an extension $\beta$-Gaussian family in the literature. Additionally, for the sake of fitting the distribution parameters to data, since gradient-based methods do not work in this scenario, they construct a wrapped Fenchel-Young loss function based on the learning objective function for $\beta$-Gaussian distribution. Next, they introduce an efficient embedded parametrization of scale matrices for the case of hyperspheres. Finally, they conducted several experiments to demonstrate the effectiveness of the wrapped $\beta$-Gaussians on many tasks, namely data fitting, generative modeling and multilingual word embedding alignment.

**Audience:**

Yes

**Claims And Evidence:**

Yes

**Requested Changes:**

1. Regarding Proposition 1, the authors should include the formulation of $A_{\beta}(\Sigma)$ in the main text. Additionally, compared to Gaussian distributions in the Euclidean space, the appearance of the term $A_{\beta}(\Sigma)$ in the exponential operator is new, so could the authors discuss more about this term?

2. Given the density formulation in Proposition 1, are we able to compute explicitly the corresponding moment generating function? If not, then what are the difficulties?

3. In equation (26), since the term $A_{\beta}(\Sigma)$ is invalid when $\beta=1$, I wonder how the authors evaluate the density function given in equation (4) when $\beta=1$.

4. In Appendix B.3, the authors compute the discrepancy between two $\beta$-Gaussian distributions of the same dimension under the Wasserstein distance. I wonder if we still have a closed-form expression for that of two $\beta$-Gaussian distributions defined on Riemannian manifolds of different dimensions.

5. Under Figure 1, the notations $\mathbb{S}^n$ and $SO(n)$ are not defined. Please add their definitions before using or call their names such as hyperspheres and special orthogonal manifolds.

6. Typos: (1) In the second paragraph of the introduction section, 'equivalent' should be 'equivalence'. (2) In Appendix B.3, 'analitical' should be 'analytical', and $WN_{\beta}(\mu_1,\Sigma_2)$ should be $\mathcal{WN}_{\beta}(\mu_1,\Sigma_1)$.

**Strengths And Weaknesses:**

**Strength**:

1. Originality: A family of wrapped $\beta$-Gaussian distributions is a novel distribution family on Riemannian manifolds which provides exact density evaluation and reparametrized sampling in high dimensions.

2. Significance: The proposed family of distributions is efficient for sampling, parameter learning and density evaluation in comparison with other distributions on Riemannian manifolds.

3. Quality: Theoretical results are solid with rigorous proof but I do not spend time double-checking all of them. Moreover, applications of wrapped $\beta$-Gaussian distributions in parameter estimation, generative modeling (variational autoencoder) and multilingual word embedding alignment.

4. Clarity: The paper is well organized and written.

**Weakness**:

1. (Statistical) bounds for the difference between parameter estimates in equations (17) and (18) and their true values are missing.
    \item A discussion on optimization of the Fenchel-Young loss function in equation (14) is missing.

 2.  Some notations have not been defined thoroughly (see Requested Changes).

---

> ### Author Response · Authors · 2023-07-19
> **Review intended for another paper?**
>
> Hi,
>
> It looks like this review is not about the current paper. I just wanted to flag this soon in order to not delay the review process of the other paper.
>
> Best,

---

> > ### Comment · Reviewer_LAzL · 2023-08-28
> > **The review was changed.**
> >
> > Dear the authors and the AE,
> >
> > Sorry for the mixup review. I just changed the review.
> >
> > Best,
> >
> > Nhat

---

> ### Author Response · Authors · 2023-09-15
>
> We thank you for the feedback and suggestions!
>
> **Requested changes 1, 3.**
> Thank you, we recognize this was not sufficiently clear, and we have updated the manuscript to improve this.
> For $\beta=1$, $A_\beta$ as defined is indeed an $\infty-\infty$ indefinite case, but the limit as $\beta \rightarrow 1$ agrees with the usual Gaussian. $1$-Gaussians are just the usual Gaussians, and can be implemented with the standard expressions.
>
> Regarding additive vs. multiplicative normalization constants, this is indeed a deeper discussion. For the Gaussian, we can write the density in two equivalent ways: 1) $p(x) = \exp(f(x)) / Z$; 2) $p(x) = \exp(f(x) - A)$, where $A = \log Z$. Interestingly, when generalizing exp/log to Tsallis counterparts, these two forms give two different constructions ($\exp_\beta(x + y) \ne \exp_\beta(x)\exp_\beta(y)$ for $ \beta \ne 1$). Both have been studied in the literature and have different properties, see Martins et al. 2022, Appendix C.4 for details. We adopt the formulation most suitable to our use case.
>
> **Requested change 2.**
> We are not aware of an expression of the moment generating function for other wrapped or intrinsic distributions on Riemannian manifolds. This is an interesting question and we would appreciate any pointers in this direction for further research. For distributions defined in ambient Euclidean space, the moments are not usually discussed in a Riemannian way, for instance, the "first moment" of the Langevin distribution is taken to be inside the sphere (Mardia and Jupp, 2009, eq 9.3.7). For statistics on manifolds, like Pennec et al (2006), we must use the intrinsic notion of moments.
>
> **Weakness 1.**
> Thank you for bringing up the insightful comment on statistical bounds for the difference between parameter estimates and true values of location and scale. We hadn’t considered this direction yet, but our preliminary research now suggest the results are favorable for wrapped $\beta$-Gaussians. Analysing statistical properties of the Fréchet mean is more challenging than the Euclidean case, but several central results are present (Bhattacharya and Patrangenaru’03, Pennec’19, Schötz’21). The uniqueness of the Fréchet mean, and the convergence guarantees of the sample mean are tightly bound to the Kendall & Karcher concentration conditions (Karcher’77 & Kendall’89): the support of the distribution must be fully inside the geodesic ball of radius $r < \frac{1}{2} \min(\text{inj}_M, \pi / \sqrt{\kappa})$, where $\kappa$ is the sectional curvature (assumed non-negative, as for negative curvature manifolds the assumption is always satisfied). Unlike the Gaussian distribution, which has infinite support, we can control the support size and thus choose parameters to satisfy the Kendall and Karcher condition.
> For the sphere, this would concretely mean constraining the support to the geodesic ball $B(\mu, \pi/2)$ rather than the ball of radius $\pi$ needed for injectivity. We will include this discussion in the parameter estimation section. We hope that this connection will strengthen the paper and provide new insights.
> As for higher moments, assuming the first moment is known, the estimation takes place in tangent (Euclidean) space, so standard results would hold.
>
> **Weakness 2.**
> Regarding your comment on Fenchel-Young loss optimization, the two Fenchel-Young losses behave similarly in our experiments (Figure 10). Equation 14 introduces a Jacobian term to account for wrapping (the same as in Equation 10), and we hope the revision clarifies the notation here. The Jacobian term has the exact expression listed in Table 1, and is efficiently computable in the manifolds we experiment with; no higher-order differentiation is needed. We hope we understood your concern regarding loss optimization or the question was referring to something else?
>
> **Requested change 4.**
> Regarding the question on the Wasserstein distance between wrapped $\beta$-gaussians defined in different spaces. We can see two possible scenarios: 1) if we have two distributions with different support but living in the same manifold M (e.g. if the covariances of the distributions are low-rank), the pullback W2 construction we give works as is; 2) if the distributions are defined on different manifolds (eg no given anchoring between them, no predefined inclusion relationship) then the Wasserstein distance does not apply, as it requires a distance between the samples of one distribution and the other. In this scenario, the Gromov-Wasserstein distance would apply. A similar “pullback” construction might be possible; this is an interesting question possibly worth studying in the future, do you have any further suggestions for where this scenario would arise?
>
> **Requested change 5, 6 and weakness 3.**
> We have fixed the mentioned typos and notation issues in the text, thanks!

---

> ### Author Response · Authors · 2023-09-15
>
> **References**
> - Karcher, H. (1977) Riemannian Center of Mass and Mollifier Smoothing. Communications on Pure and Applied Mathematics, 30, 509-541.https://doi.org/10.1002/cpa.3160300502
> - David G. Kendall. "A Survey of the Statistical Theory of Shape." Statist. Sci. 4 (2) 87 - 99, May, 1989. https://doi.org/10.1214/ss/1177012582
> - Rabi Bhattacharya. Vic Patrangenaru. "Large sample theory of intrinsic and extrinsic sample means on manifolds." Ann. Statist. 31 (1) 1 - 29, Februrary 2003. https://doi.org/10.1214/aos/1046294456
> - Pennec, Xavier. (2019). Curvature effects on the empirical mean in Riemannian and affine Manifolds: a non-asymptotic high concentration expansion in the small-sample regime.
> - André F. T. Martins et al.. Sparse continuous distributions and Fenchel-Young losses. Journal of Machine Learning Research, 23(257):1–74, 2022. URLhttp://jmlr.org/papers/v23/21-0879.html

---

### Review · Reviewer_NTsP · 2023-07-24

**Summary Of Contributions:**

The paper considers wrapped $\beta$-Gaussians on embedded spheres and rotation matrices. This generalizes a compactly-supported generalization of Gaussians to these manifolds, allowing for efficient sampling and exact density evaluation. Furthermore, they provide Fenchel-Young losses, which are $\beta$-generalized analogues of the KL divergence and cross-entropy loss. Proof-of-principle experiments are done with moment matching, hierarchy modelling, VAEs, and multilingual embedding alignment.

**Audience:**

Yes

**Broader Impact Concerns:**

I do not have any concerns in this direction.

**Claims And Evidence:**

Yes

**Requested Changes:**

I feel the work is just above the bar, due to its careful theoretical contributions on (narrow-ish) domains of interest. The work would be significantly strengthened by addressing the two weaknesses I mentioned above. The first may be dealt with via some rephrasing and limitation acknowlegement in the introduction. The second is more challenging, as it involves additional experiments, and perhaps example applications that are entirely novel. Without a clear application that shows the advantages of using these wrapped $\beta$-Gaussians, it becomes less likely that others in the community will consider them for use.

**Strengths And Weaknesses:**

Strengths:
- The exposition and organization is clear, with precise statements, intuition explained, and proofs and details pushed to appendices.
- Much work has been put into coming up with usable algorithms in the sphere and rotation manifold cases, ensuring that distribution supports are within the injectivity radius and incorporating the nontrivial exponential/logarithm/parallel transport maps.

Weaknesses:
- The method perhaps overclaims in its generality, as utility of the method requires analytical exponential, logarithm, and parallel transport maps. This is a small class of manifolds, as illustrated with just two example (classes) in their experiments.
- It could be argued that the experimental results are relatively weak.
   - The first experiment on moment matching is merely a proof-of-concept, and one would typically use simple sampling to determine these quantities.
   - The second experiment is quite anecdotal, and there is not commentary on whether this method improves upon previous embeddings into Euclidean space.
   - The third experiment only outperforms the baseline in lower dimensions, and the increase in performance seems relatively small.
   - The fourth experiment again does slightly worse than the baseline, so merely serves as an example that the basic idea works for $SO(n)$.

---

> ### Author Response · Authors · 2023-09-15
> **Respond:**
>
> We thank the reviewer for the feedback and suggestions!
>
> **Weakness 1 (Generality of the method).**
> We agree with your concern that we might be overstating the generality. We have added the discussion on the generality of our method to the limitation section, and we shall clarify this further in the next revision. We do believe this depends slightly on perspective, as tractable $Exp$ and $Log$ mappings are often required in Riemannian optimization, and we are used with working with manifolds for which these are available. For example, we could also apply our construction to torii (and other product manifolds), other matrix manifolds such as $SE(n) $(e.g., all cases discussed in Chevallier et al ’22), and of course negative-curvature manifolds like the Lorentz hyperbolic model. Our construction has similar requirements to any wrapped distribution, and we would argue is more user-friendly than the intrinsic approach, which requires derivations of normalizing constants and sampling as in Hauberg’18 for each new manifold.
>
> We also point out that Log doesn't need to be implemented unless we need to assess densities $p(y)$ at arbitrary points. For instance, in the Bayesian Procrustes experiment we only assess densities at samples drawn from the posterior approximation, so we did not need to implement Log.
>
> **Weakness 2 (Limited experiments).**
> Despite our admittedly limited experiments, we hope we were able to demonstrate with a proof of concept the applicability of our construction, where we can train anisotropic probabilistic models effectively even in high-dimension spaces. We also hope that our code (to be open-sourced) can facilitate further research in this direction and enable larger-scale experiments.
>
> **References**.
> - Chevallier, E., Li, D., Lu, Y., & Dunson, D. (2022). Exponential-Wrapped Distributions on Symmetric Spaces. SIAM Journal on Mathematics of Data Science, 4(4), 1347-1368.

---

### Review · Reviewer_L93q · 2023-09-04

**Summary Of Contributions:**

The paper is concerned with tractable distributions on Riemannian manifolds. The main idea is to consider $\beta$-Gaussians, which are Euclidean distributions with compact support. These distributions can then be placed in the tangent space of a manifold. As long as the compact distribution is fully supported within the injectivity radius, you avoid all the difficulties of wrapping on manifolds.

This is rather nice, I think.

This is one of those papers that are obvious if you know both about $\beta$-Gaussians and manifold statistics. In practice, most readers will not be familiar with both (I was not familiar with $\beta$-Gaussians, for example), and the paper provides a very nice contribution to such readers.

All in all, I recommend acceptance of the manuscript subject to the requested changes being made.

**Audience:**

Yes

**Broader Impact Concerns:**

The paper is on the 'theory side', so I do not see any ethical issues.

**Claims And Evidence:**

Yes

**Requested Changes:**

It would be good to fix the weaknesses above (should all be easily done).

**Strengths And Weaknesses:**

## Strengths
* The paper makes a very clear contribution, which is practically useful for manifold statistics.
* The paper is well-written and generally easy to follow.
* The paper is fairly complete (I do not miss anything significant from the paper as is).
* References are generally appropriate.

## Weaknesses
* In general, I miss illustrative plots. Specifically, I would like to see plots of (to guide intuitions)
  * The $\beta$-Gaussian density for different values of $\beta$.
  * The wrapped $\beta$-Gaussian on a sphere for different values of $\beta$.
  * The functions $\exp_{\beta}$ and $\log_{\beta}$
* I miss a discussion of the limitations of the model. At least, I see that the model is quite limited on manifolds with a small injectivity radius. Also, distributions defined in tangent spaces often end up behaving quite differently when mapped to the manifold than one would expect (the tangent space can give a highly distorted view of the manifold).

## Minor comments
* On page 3, Chevallier & Guigui are cited as the creators of 'The wrapped approach', which strikes me as misleading. That's a much older strategy; I think the book by Mardia & Jupp discusses wrapped Gaussians, for example.
* 'FY-losses' does not appear to be defined (I can guess the meaning, but it should be defined nonetheless).
* In Eq. 2, the notation $[\cdot]_+$ is not defined. I assume it's the usual hinge $\max(\cdot, 0)$, but it should be defined.

## Questions
* Is the wrapped $\beta$-Gaussian density on the sphere continuous and/or differentiable at the antipodal point?
* As far as I could tell, the wrapped $\beta$-Gaussian is used with VAEs to give the latent space a pre-specified geometric structure. Is it easy to use the same distribution as the conditional likelihood $p(x|z)$?

---

> ### Author Response · Authors · 2023-09-15
> **Response**
>
> We thank you for the feedback and suggestions!
>
> **Weaknesses.**
> We have added the requested plots and a limitation section, we agree these are an improvement.
>
> **Question 1 (Antipodal point).**
> In our construction on the sphere, the support lies entirely within a geodesic ball of radius less than $\pi$. The antipodal point therefore has zero density, since its preimage in the tangent space has zero density. We also add that the density of a WBG is continuous and differentiable inside the support; however, depending on $\beta$, it might not be differentiable on the boundary of the support.
>
> **Question 2 (VAE).**
> Indeed, in the context of VAE, wrapped $\beta$ Gaussian is only used to model the latent distribution. Using sparse support distributions for the observation model $p(x|z)$ is more challenging, as log-probabilities can be infinite. One could replace $p(x|z)$ in ELBO with Fenchel-Young loss, but unfortunately, the connection to ELBO does not hold. This could potentially lead to VAE-inspired learning objective, but not exactly VAE.
>
> **Minor comments.**
> Regarding historical citations for wrapped distributions: this is a good point, the approach is well-known, and we agree we should provide some historical and not only modern comprehensive work. We replace it by citing Mardia and Jupp, 1999, and Stephens, 1963. We add the proper mention of FY losses in the text and improve the structure of this section.
> We add the definition of $[]_+$ in the text.
>
> **References**
> - Michael. A. Stephens. Random walk on a circle. Biometrika, 50(3-4):385–390, 12 1963. ISSN 0006-3444. doi:10.1093/biomet/50.3-4.385. URL https://doi.org/10.1093/biomet/50.3-4.385.

---

> > ### Comment · Reviewer_L93q · 2023-09-15
> > **Thank you**
> >
> > I am pleased with the follow up reply from the authors and I will be recommending acceptance of the paper.

---

### Author Response · Authors · 2023-09-15
**Revision submitted**

We thank all the reviewers for their helpful feedback! We have incorporated most of the required changes in the current revision, and plan to submit another text revision at the end of the discussion period. We have exceeded a bit the 12-page limit to incorporate the requested changes, we hope this doesn't complicate the submission process, otherwise, we will try to make it to the 12-page limit.

---

> ### Author Response · Authors · 2023-09-18
> **Revision submitted**
>
> Dear reviewers, we have finalized the paper revision.
> Changes after discussion period:
> - rewrote paragraph on Fenchel-Young losses improving readability, added $A_\beta$ definition;
> - added discussion on $A_\beta$ term and relation to Gaussian distribution for $\beta=1$;
> - added discussion on moments estimation in paragraph 5.1;
> - added limitation section: distortions, manifolds with small injectivity radius, generality;
> - added discussion on the generality of WBG to the introduction and the limitation section;
> - added new visualizations to help the reader;
> - fixed typos

---

### Decision · Action_Editor_CB9c · 2023-12-01

**Recommendation:** Accept as is

**Comment:**

The support for the paper was strong overall, and the authors were satisfied with the rebuttal.

**Audience:**

The contribution of this paper is quite fundamental and low-level, in the sense that it could pop up again as a basic block in various applications. Therefore I believe the audience will be quite wide, including of course statisticians with a taste for geometric data.

**Claims And Evidence:**

This paper proposes a new probability density model for Riemannian manifolds with explicitly (known) log/exp maps. The work extends the recently proposed framework of beta-Gaussians using the wrapped densities machinery recently highlighted by Chevallier (2022). I enjoyed reading the paper, and so did all reviewers: the content is very clearly introduced, the visualizations are very well done, and quite beautiful, and the experiments, while still a bit toyish, and revisiting the older task of word embeddings, are convincing enough as companions to this otherwise novel method.